# A  Appendix

## A.1  TensorFlow Primitives Vocabulary

| Name | TF Function | Argument Mapping | | | |
|---|---|---|---|---|---|
| | | Input 1 | Input 2 | Constant | Dim Size |
| ADD | tf.math.add | x | y | - | - |
| DIFFERENCE | tf.math.subtract | x | y | - | - |
| DIVIDE | tf.math.divide | x | y | - | - |
| MULTIPLY | tf.math.multiply | x | y | - | - |
| ABS ROOT | tf.math.sqrt(tf.abs(x)) | x | - | - | - |
| SQUARE | tf.math.square | x | - | - | - |
| EXP | tf.exp | x | - | - | - |
| LOG | tf.log(tf.abs(x)) | x | - | - | - |
| C MUL | tf.math.multiply | x | - | y | - |
| ABS | tf.abs | x | - | - | - |
| RECIP | tf.math.reciprocal_no_nan | x | - | - | - |
| SIGN | tf.sign | x | - | - | - |
| COS | tf.cos | x | - | - | - |
| SIN | tf.sin | x | - | - | - |
| TANH | tf.tanh | x | - | - | - |
| MAX | tf.math.maximum | x | - | y | - |
| MIN | tf.math.minimum | x | - | y | - |
| SCALE | x+tf.Variable() | x | - | - | - |
| SHIFT | x*tf.Variable() | x | - | - | - |
| SIGMOID | tf.sigmoid | x | - | - | - |
| MASK | tf.linalg.band_part | input | - | - | - |
| CUM PROD | tf.math.cumprod | x | - | | - |
| CUM SUM | tf.math.cumsum | x | - | - | - |
| RED MEAN | tf.reduce_mean | input_tensor | - | - | - |
| RED SUM | tf.reduce_sum | input_tensor | - | - | - |
| RED MIN | tf.reduce_min | input_tensor | - | - | - |
| RED MAX | tf.reduce_max | input_tensor | - | - | - |
| RED PROD | tf.reduce_prod | input_tensor | - | - | - |
| MAT MUL | tf.matmul | a | b | - | - |
| T-MAT MUL | tf.matmul(transpose_b=True) | a | b | - | - |
| CONV 1X1 | tf.layers.dense | inputs | - | - | units |
| CONV 3X1 | tf.nn.conv1d | input | - | - | filters |
| CONV 7X1 | tf.nn.conv1d | input | - | - | filters |
| CONV 15X1 | tf.nn.conv1d | input | - | - | filters |
| CONV 31X1 | tf.nn.conv1d | input | - | - | filters |
| DCONV 3X1 | tf.nn.depthwise_conv2d | input | - | - | filters |
| DCONV 7X1 | tf.nn.depthwise_conv2d | input | - | - | filters |
| DCONV 15X1 | tf.nn.depthwise_conv2d | input | - | - | filters |
| DCONV 31X1 | tf.nn.depthwise_conv2d | input | - | - | filters |

Table 2: TensorFlow (TF) Primitives Vocabulary. "Name" is the name of the operation in our search space. "TF Function" is the TensorFlow function that the name is mapped to when a DNA instruction is being converted to a line of TensorFlow code. "Argument Mapping" describes how the values in a DNA's argument set are mapped to the corresponding TensorFlow function arguments. This vocabulary is largely constructed from the lowest level TF operations needed to create Transformers (see Appendix A.5). We additionally extend those operations to include adjacent operations; for example, we extend MAX to also include MIN, extend RED SUM to include RED PRODUCT, and extend CONV 1X1 to include CONV 3X1. We also add commonly used math primitives such as SIN and ABS.

## A.2   Constructing TensorFlow Graphs

TensorFlow graphs are built from DNA programs as described in Section 2 of the main text. Here we provide additional implementation details.

**Relative Dimensions:**   We use relative dimensions [13] instead of absolute dimensions for each instruction's "dimension size" argument. This allows us to resize the models to fit within our parameter limits (32M to 38M parameters). The vocabulary for these relative dimensions is [1, 2, 4, 8, 12, 16, 24, 32, 48, 64]. This vocabulary was not tuned.

**Values Bank:**   For "constant" and "dimension size" argument fields, we create a shared bank of values that each instruction references. The constants bank holds 2 values and the dimension sizes bank holds 6 values; these numbers were not tuned. Instead of each instruction possessing their own individual values for these arguments, they instead hold an index to these shared banks. This allows multiple instructions to share the same value and to change simultaneously when that value is changed. For example, each of the individual attention multi-head projections for $Q$, $K$ and $V$ start off sharing the same output dimension size so that they all change simultaneously if that value changes. See A.4 for an example of how these bank values are mutated.

**Causal Masking:**   An important part of teacher-forced language model training is that positions cannot "see" the token they are trying to predict. Each position should only get information from previous positions, otherwise the model will be degenerate when the targets are not provided. To enforce this causal constraint we add additional overhead to operations that move information spatially to mask out any information from future positions. For example, when applying convolutions we follow the standard practice of shifting the inputs spatially by (KERNEL WIDTH − 1) so that each position only receives information from previous positions.

**Branching:**   To enable multi-head capabilities for the Transformer search seed, we add a meta argument to our instructions called "branching." This argument can take any value in [1, 2, 4, 8, 16] and determines how many times that instruction is executed in parallel, with the resulting tensors being concatenated together along their embedding axes. Branching can be used with any of the TensorFlow primitives as well as with any of a DNA's subprograms. This allows us to initialize the search with multi-head self-attention by branching SUBPROGRAM 1 (self-attention) 8 times (see Appendix A.5 for subprogram implementations). Primer does not utilize this branching capability in any meaningful way, beyond using the initialized multi-head attention.

**Resolving Dimension Mismatches:**   We do not constrain how tensor dimensions can be mutated and so programs may be invalid because they perform binary operations on tensors with incompatible sizes. For example, a program may describe adding together two tensors with differing embedding sizes. To resolve these dimension mismatch issues we deterministically pseudorandomly set one of the tensor dimensions to match the other.

## A.3   Halving Hurdles

We configure our hurdles [13] such that the top 50% of individuals passes each hurdle, according to fitness. We space the hurdles in such a way that the expected amount of compute devoted to training each hurdle band is roughly equal at the end of the search. That is, given that our maximum amount of training compute for an individual is 7 hours or 25,200 seconds (s), we construct hurdles at the 812.9s, 2438.7s, 5690.3s, and 12,193.5s marks. Thus, 1/5 of the compute budget is devoted to training every individual up to the first hurdle (812.9s), 1/5 of the compute budget is devoted to training the ∼50% of individuals that are trained from the first to the second hurdle (2438.7s − 812.9s = 1625.8s), 1/5 of the compute budget is devoted to training the ∼25% of individuals that are trained from the second to the third hurdle (5690.3s − 2438.7s = 3251.6s), etc. This configuration strategy, which we refer to as "halving hurdles," requires setting only one hyperparameter, the number of hurdles, and removes the need to set hurdle threshold values and comparison steps, as has been previously done [13, 35]. We choose four hurdles because five hurdles would require the first hurdle to be anchored at less than ten minutes of training, which we find empirically to be too noisy of a signal. Using hurdles in this way decreases the average train time per model to 4064s or about 1 hour and 8 minutes, reducing the compute cost by a factor of ∼6.2X.

This strategy is not unlike bandit algorithms such as Successive Halving[53] and Hyperband[54], however we do not use a static population of individuals created a priori, but integrate our halving with the changing evolutionary population.

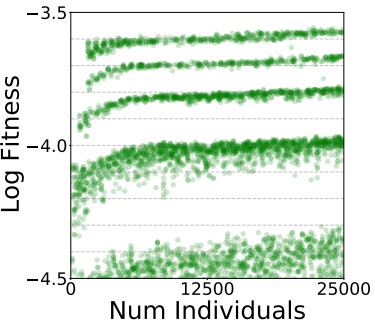

Figure 12: Halving hurdles from our Primer search. Each dot represents the final fitness of an individual generated by evolution. Different "bands" form because each hurdle has a different training allowance. All bands see improvement over time, meaning that the median fitness improves for all compute allowances. This correlation between a model's performances at different training budgets allows us to reduce our total search cost by roughly a factor of 6.2X.

## A.4 Evolution Search Details

We use Regularized Evolution [30] with a population size of 100 and a tournament selection size of 10. These values were not tuned. The mutations we use are as follows.

**Mutations:** To create new candidates in our search, we uniform randomly select a *parent* from our search population and apply a single *mutation* to it. We employ five different mutation types (selections and decisions are performed uniform randomly unless specified otherwise):

- *Delete*: Remove an instruction from a subprogram.
- *Insert*: Create an instruction and insert it into a subprogram.
- *Delete and Insert*: Perform a delete mutation followed by an insert mutation [55].
- *Mutate Field*: Select a field from an instruction and change its value.
- *Swap*: Swap the position of two instructions in a randomly selected subprogram. The input tensors for each instruction are also swapped so that the net effect is switching the positions of the instructions in the compute graph.
- *Mutate Bank Value*: Change the value of a relative tensor dimension or constant in the corresponding bank. The values for relative tensor dimensions are selected from their vocabulary (see Appendix A.2). The values for constants are changed according to $c_{new} := c_{prev} \cdot 10^X + Y$ for previous value $c_{prev}$, new value $c_{new}$ and random variables $X, Y \sim N(0, 1)$.

After a mutation is applied, we run a light check to see if the resulting candidate's compute graph is exactly equivalent to the parent's compute graph. If it is, we perform another mutation.

## A.5 Transformer and Primer Program Comparisons

Here we present the programs for both the Transformer seed and the discovered Primer model. Table 3 is a key that maps operation names to graph symbols for subsequent graphs. Figures 13 to 22 depict the subprograms for each model with the Primer changes highlighted in orange. Figure 23 depicts the full compute graphs for each model, with all subprograms resolved to their constituent primitives. Figures 24 and 25 depict the DNA programs for Transformer and Primer with all subprograms resolved and all instruction bank values plugged in.

| Name | Graphing symbol |
|------|-----------------|
| ADD | $+$ |
| DIFFERENCE | $-$ |
| DIVIDE | $\div$ |
| MULTIPLY | $\times$ |
| ABS ROOT | $\sqrt{}$ |
| SQUARE | $x^2$ |
| EXP | $e^x$ |
| LOG | Log |
| C MUL | $\times C$ |
| ABS | $\mid x \mid$ |
| RECIP | Recip |
| SIGN | Sign |
| COS | Cos |
| SIN | Sin |
| TANH | Tanh |
| MAX | Max |
| MIN | Min |
| SCALE | Scale |
| SHIFT | Shift |
| SIGMOID | Sigm |
| MASK | Mask |
| CUM PROD | Cum Prod |
| CUM SUM | Cum Sum |
| RED MEAN | Reduce Mean |
| RED SUM | Reduce Sum |
| RED MIN | Reduce Min |
| RED MAX | Reduce Max |
| RED PROD | Reduce Prod |
| MAT MUL | $M \times N$ |
| T-MAT MUL | $M \times N^T$ |
| CONV 1X1 | Conv 1x1 |
| CONV 3X1 | Conv 3x1 |
| CONV 7X1 | Conv 7x1 |
| CONV 15X1 | Conv 15x1 |
| CONV 31X1 | Conv 31x1 |
| DCONV 3X1 | D-wise 3x1 |
| DCONV 7X1 | D-wise 7x1 |
| DCONV 15X1 | D-wise 15x1 |
| DCONV 31X1 | D-wise 31x1 |

Table 3: Key for primitives mapped to corresponding symbols used in the following graphs.

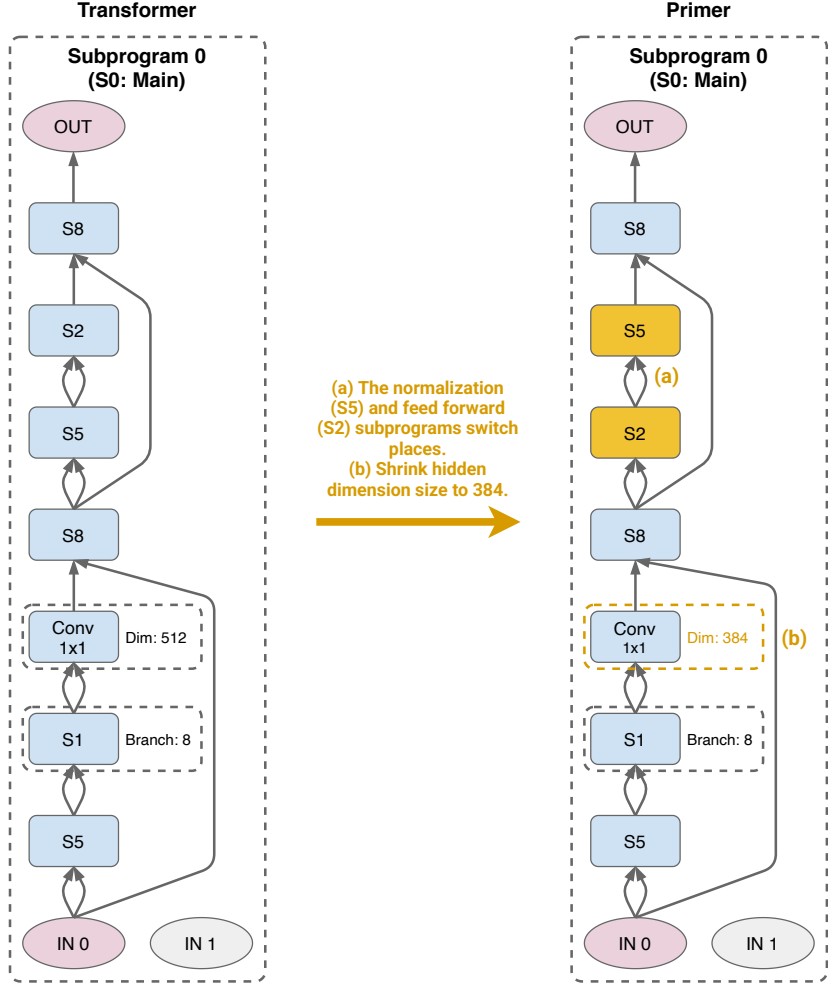

Figure 13: Main subprograms. Changes are highlighted in orange.

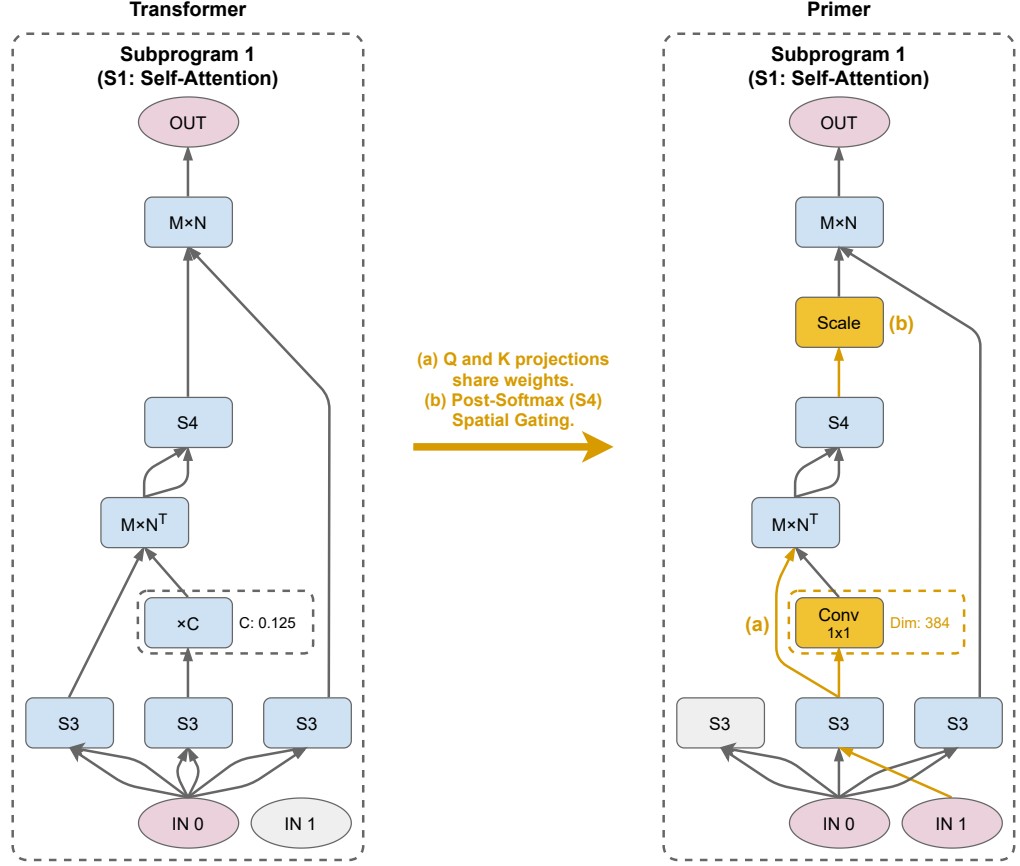

Figure 14: Attention subprograms. Changes are highlighted in orange.

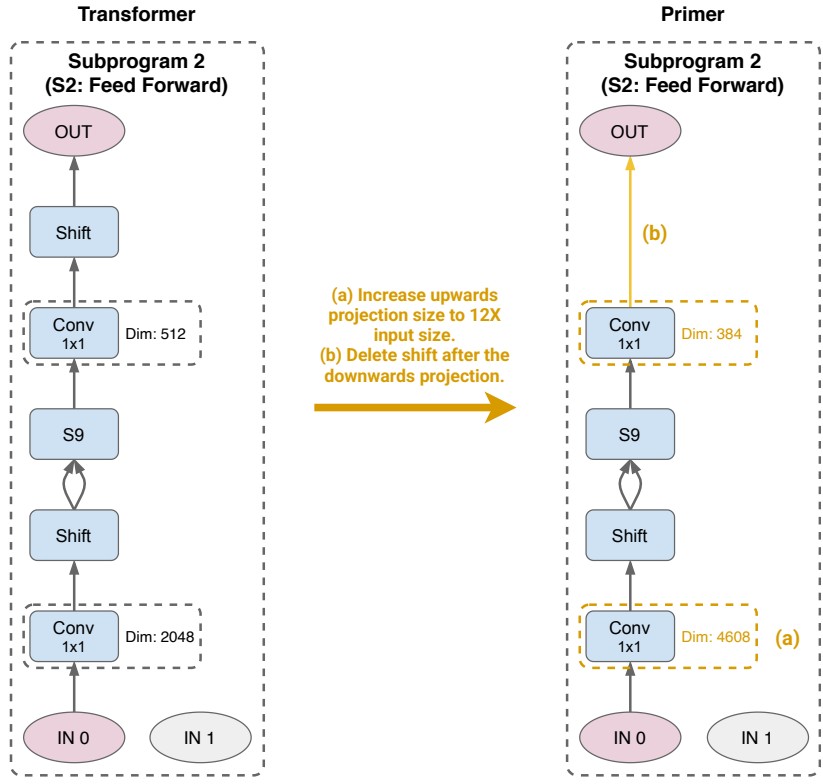

Figure 15: Feed forward subprograms. Changes are highlighted in orange.

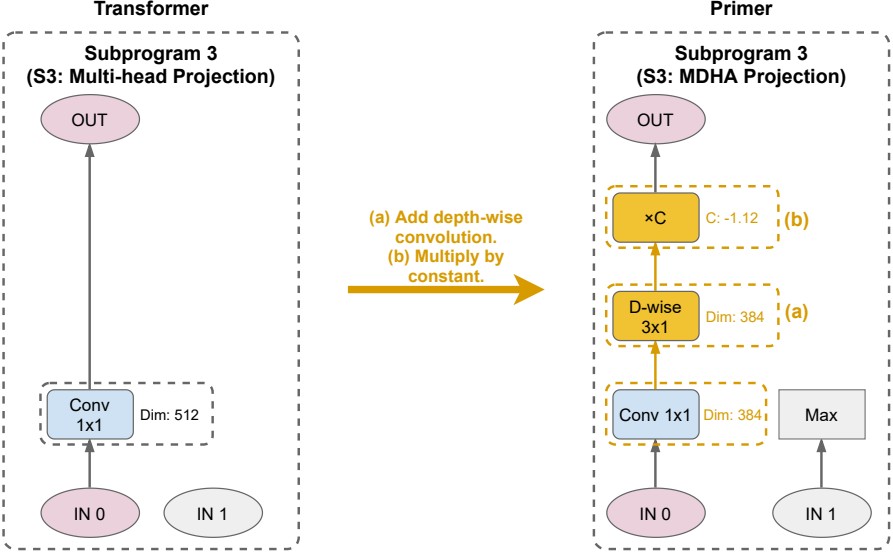

Figure 16: Multi-head projection subprograms. Changes are highlighted in orange.

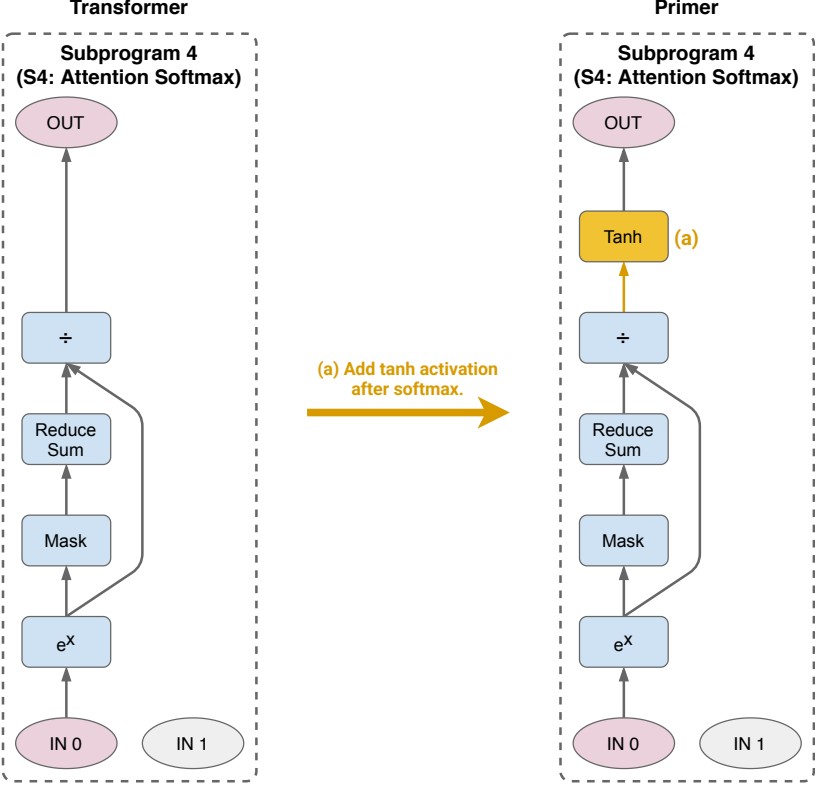

Figure 17: Softmax subprograms. Changes are highlighted in orange.

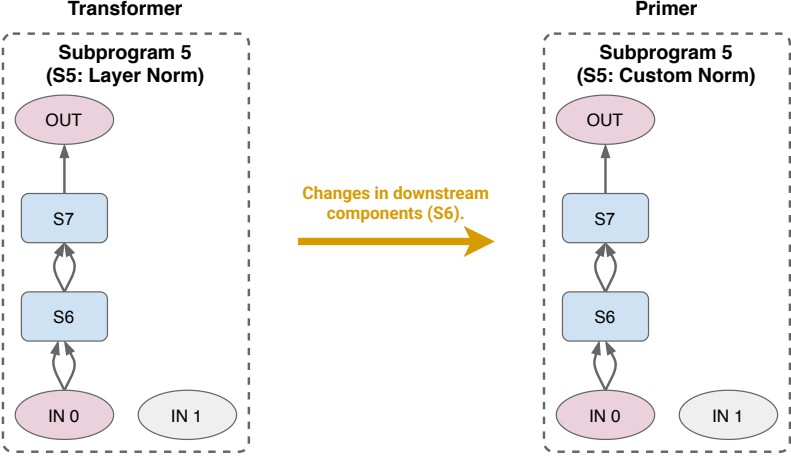

Figure 18: Normalization subprograms. Changes to this subprogram are realized in downstream changes to S6.

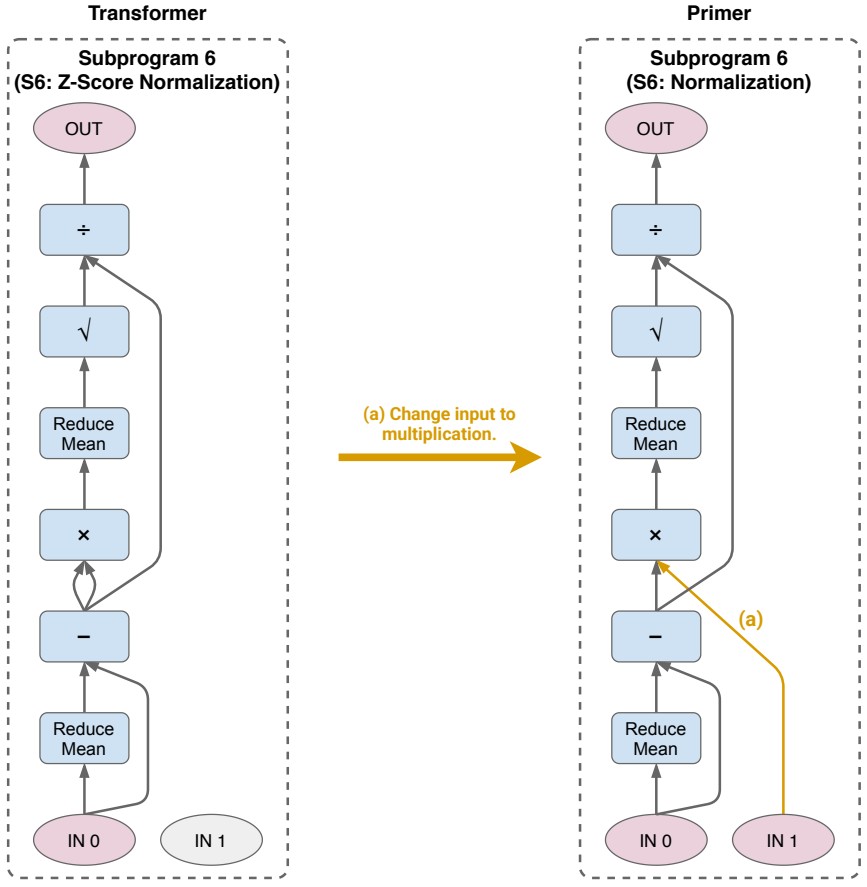

Figure 19: Z-score normalization subprograms. Changes are highlighted in orange.

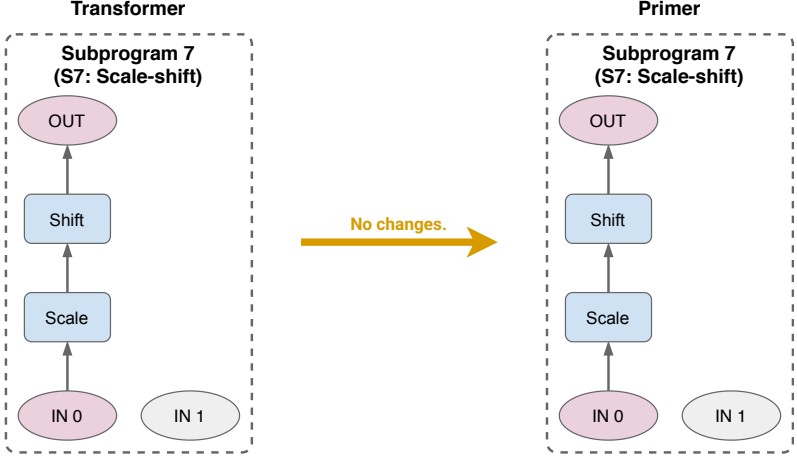

Figure 20: Scale-shift subprograms. No changes here.

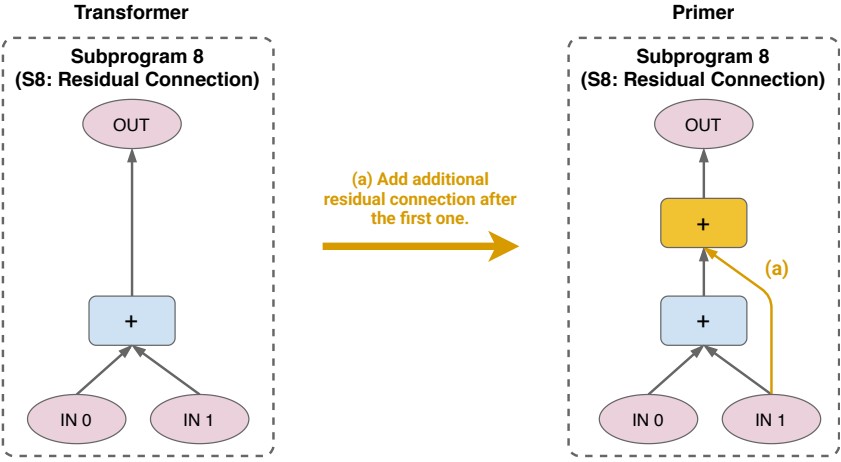

Figure 21: Residual connection subprograms. This change is essentially a functional no-op.

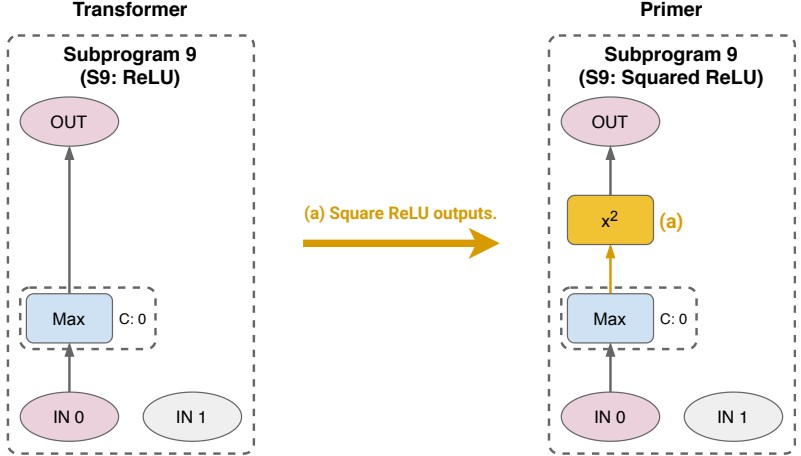

Figure 22: Activation function subprograms. Changes are highlighted in orange.

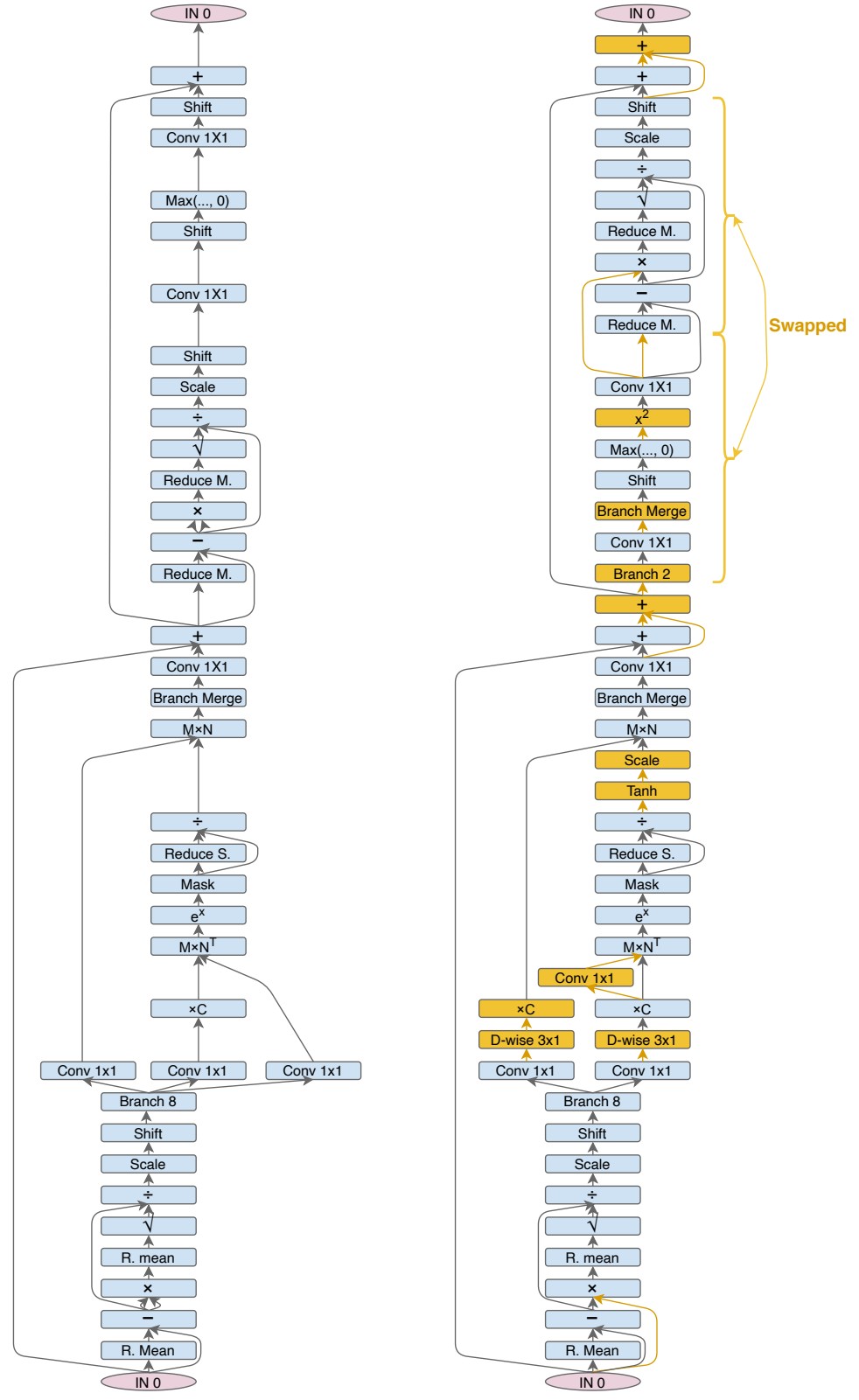

Figure 23: Comparison of Transformer (Left) and Primer (Right) programs, with all subprograms resolved to their constituent primitives. Primer differences are highlighted in orange.

```
                              TRANSFORMER

(0)   INPUT
(1)   INPUT
(2)   REDUCE_MEAN          In0: 0    In1: 0    Dim: 128    C: 0.00
(3)   DIFFERENCE           In0: 0    In1: 2    Dim: 128    C: 0.00
(4)   MULTIPLY             In0: 3    In1: 3    Dim: 128    C: 0.00
(5)   REDUCE_MEAN          In0: 4    In1: 4    Dim: 128    C: 0.00
(6)   ABS_SQUARE_ROOT      In0: 5    In1: 5    Dim: 128    C: 0.00
(7)   DIVIDE               In0: 3    In1: 6    Dim: 128    C: 0.00
(8)   SCALE                In0: 7    In1: 7    Dim: 128    C: 0.00
(9)   SHIFT                In0: 8    In1: 8    Dim: 128    C: 0.00
(10)  BRANCH_8_INPUT_1     In0: 9    In1: 9    Dim: 128    C: 0.00
(11)  BRANCH_8_INPUT_2     In0: 9    In1: 9    Dim: 128    C: 0.00
(12)  DENSE                In0: 10   In1: 10   Dim: 64     C: 0.00
(13)  DENSE                In0: 10   In1: 10   Dim: 64     C: 0.00
(14)  CONSTANT_MUL         In0: 13   In1: 13   Dim: 128    C: 0.12
(15)  DENSE                In0: 10   In1: 10   Dim: 64     C: 0.00
(16)  TRANSPOSE_MAT_MUL    In0: 14   In1: 12   Dim: 128    C: 0.00
(17)  EXP                  In0: 16   In1: 16   Dim: 128    C: 0.00
(18)  EMBEDDING_MASK       In0: 17   In1: 17   Dim: 128    C: 0.00
(19)  REDUCE_SUM           In0: 18   In1: 18   Dim: 128    C: 0.00
(20)  DIVIDE               In0: 18   In1: 19   Dim: 128    C: 0.00
(21)  MAT_MUL              In0: 20   In1: 15   Dim: 128    C: 0.00
(22)  BRANCH_MERGE         In0: 21   In1: 21   Dim: 512    C: 0.00
(23)  DENSE                In0: 22   In1: 22   Dim: 512    C: 0.00
(24)  ADD                  In0: 0    In1: 23   Dim: 128    C: 0.00
(25)  REDUCE_MEAN          In0: 24   In1: 24   Dim: 128    C: 0.00
(26)  DIFFERENCE           In0: 24   In1: 25   Dim: 128    C: 0.00
(27)  MULTIPLY             In0: 26   In1: 26   Dim: 128    C: 0.00
(28)  REDUCE_MEAN          In0: 27   In1: 27   Dim: 128    C: 0.00
(29)  ABS_SQUARE_ROOT      In0: 28   In1: 28   Dim: 128    C: 0.00
(30)  DIVIDE               In0: 26   In1: 29   Dim: 128    C: 0.00
(31)  SCALE                In0: 30   In1: 30   Dim: 128    C: 0.00
(32)  SHIFT                In0: 31   In1: 31   Dim: 128    C: 0.00
(33)  DENSE                In0: 32   In1: 32   Dim: 2048   C: 0.00
(34)  SHIFT                In0: 33   In1: 33   Dim: 128    C: 0.00
(35)  MAX                  In0: 34   In1: 34   Dim: 128    C: 0.00
(36)  DENSE                In0: 35   In1: 35   Dim: 512    C: 0.00
(37)  SHIFT                In0: 36   In1: 36   Dim: 128    C: 0.00
(38)  ADD                  In0: 24   In1: 37   Dim: 128    C: 0.00
```

Figure 24: List of instructions for Transformer program, with all subprograms resolved to their constituent primitives.

```
                                          PRIMER

(0)    INPUT
(1)    INPUT
(2)    REDUCE_MEAN           In0: 0    In1: 0    Dim: 768    C: -1.12
(3)    DIFFERENCE            In0: 0    In1: 2    Dim: 768    C: -1.12
(4)    MULTIPLY             In0: 3    In1: 0    Dim: 768    C: -1.12
(5)    REDUCE_MEAN          In0: 4    In1: 4    Dim: 768    C: -1.12
(6)    ABS_SQUARE_ROOT      In0: 5    In1: 5    Dim: 768    C: -1.12
(7)    DIVIDE               In0: 3    In1: 6    Dim: 768    C: -1.12
(8)    SCALE                In0: 7    In1: 7    Dim: 768    C: -1.12
(9)    SHIFT                In0: 8    In1: 8    Dim: 384    C: -0.57
(10)   BRANCH_8_INPUT_1     In0: 9    In1: 9    Dim: 768    C: -1.12
(11)   BRANCH_8_INPUT_2     In0: 9    In1: 9    Dim: 768    C: -1.12
(12)   MAX                  In0: 10   In1: 10   Dim: 768    C: -0.57
(13)   DENSE                In0: 10   In1: 10   Dim: 48     C: -1.12
(14)   DEPTHWISE_CONV_3X1   In0: 13   In1: 10   Dim: 384    C: -1.12
(15)   CONSTANT_MUL         In0: 14   In1: 14   Dim: 384    C: -1.12
(16)   MAX                  In0: 11   In1: 11   Dim: 768    C: -0.57
(17)   DENSE                In0: 10   In1: 10   Dim: 48     C: -1.12
(18)   DEPTHWISE_CONV_3X1   In0: 17   In1: 10   Dim: 384    C: -1.12
(19)   CONSTANT_MUL         In0: 18   In1: 18   Dim: 384    C: -1.12
(20)   DENSE                In0: 19   In1: 11   Dim: 48     C: -1.12
(21)   MAX                  In0: 10   In1: 10   Dim: 768    C: -0.57
(22)   DENSE                In0: 10   In1: 10   Dim: 48     C: -1.12
(23)   DEPTHWISE_CONV_3X1   In0: 22   In1: 10   Dim: 384    C: -1.12
(24)   CONSTANT_MUL         In0: 23   In1: 23   Dim: 384    C: -1.12
(25)   TRANSPOSE_MAT_MUL    In0: 20   In1: 19   Dim: 768    C: -1.12
(26)   EXP                  In0: 25   In1: 25   Dim: 768    C: -1.12
(27)   EMBEDDING_MASK       In0: 26   In1: 26   Dim: 768    C: -1.12
(28)   REDUCE_SUM           In0: 27   In1: 27   Dim: 768    C: -1.12
(29)   DIVIDE               In0: 27   In1: 28   Dim: 768    C: -1.12
(30)   TANH                 In0: 29   In1: 25   Dim: 384    C: -1.12
(31)   SCALE                In0: 30   In1: 19   Dim: 384    C: -1.12
(32)   MAT_MUL              In0: 31   In1: 24   Dim: 768    C: -1.12
(33)   BRANCH_MERGE         In0: 32   In1: 32   Dim: 384    C: -1.12
(34)   DENSE                In0: 33   In1: 33   Dim: 384    C: -1.12
(35)   ADD                  In0: 0    In1: 34   Dim: 768    C: -1.12
(36)   ADD                  In0: 35   In1: 34   Dim: 768    C: -1.12
(37)   BRANCH_2_INPUT_1     In0: 36   In1: 36   Dim: 2304   C: -1.12
(38)   BRANCH_2_INPUT_2     In0: 36   In1: 36   Dim: 2304   C: -1.12
(39)   DENSE                In0: 37   In1: 38   Dim: 2304   C: -1.12
(40)   BRANCH_MERGE         In0: 39   In1: 39   Dim: 4608   C: -1.12
(41)   SHIFT                In0: 40   In1: 40   Dim: 768    C: -1.12
(42)   MAX                  In0: 41   In1: 41   Dim: 768    C: -0.57
(43)   SQUARE               In0: 42   In1: 41   Dim: 768    C: -1.12
(44)   DENSE                In0: 43   In1: 43   Dim: 384    C: -1.12
(45)   REDUCE_MEAN          In0: 44   In1: 44   Dim: 768    C: -1.12
(46)   DIFFERENCE           In0: 44   In1: 45   Dim: 768    C: -1.12
(47)   MULTIPLY             In0: 46   In1: 44   Dim: 768    C: -1.12
(48)   REDUCE_MEAN          In0: 47   In1: 47   Dim: 768    C: -1.12
(49)   ABS_SQUARE_ROOT      In0: 48   In1: 48   Dim: 768    C: -1.12
(50)   DIVIDE               In0: 46   In1: 49   Dim: 768    C: -1.12
(51)   SCALE                In0: 50   In1: 50   Dim: 768    C: -1.12
(52)   SHIFT                In0: 51   In1: 51   Dim: 384    C: -0.57
(53)   ADD                  In0: 36   In1: 52   Dim: 768    C: -1.12
(54)   ADD                  In0: 53   In1: 52   Dim: 768    C: -1.12
```

Figure 25: List of instructions for Primer program, with all subprograms resolved to their constituent primitives.

## A.6 Exact LM1B Numbers

| Model | Params | Train Steps | Step/Sec | PPLX | Speedup |
|---|---|---|---|---|---|
| | *Tensor2Tensor, TPUv2* | | | | |
| Vanilla Transformer | 35M | 1.9M | 22.4 | 35.44 +/- 0.30 | - |
| Transformer+GELU | 35M | 1.9M | 22.4 | 35.00 +/- 0.12 | 1.23 +/- 0.07 |
| Transformer++ | 35M | 1.9M | 22.0 | 34.87 +/- 0.46 | 1.37 +/- 0.24 |
| Primer | 34M | 1.9M | 21.7 | 33.77 +/- 0.15 | 2.12 +/- 0.09 |
| Primer-EZ | 35M | 1.8M | 21.0 | **33.53 +/- 0.09** | **2.34 +/- 0.04** |
| | | | | | |
| Transformer+MDHA | 35M | 1.8M | 21.0 | 34.26 +/- 0.12 | 1.76 +/- 0.06 |
| Transformer+Sep Conv | 35M | 1.8M | 21.0 | 34.34 +/- 0.10 | 1.54 +/- 0.05 |
| | *Tensor2Tensor, V100* | | | | |
| Vanilla Transformer | 35M | 1.3M | 15.4 | 37.19 +/- 0.07 | - |
| Transformer+GELU | 35M | 1.2M | 14.1 | 37.11 +/- 0.02 | 1.05 +/- 0.02 |
| Transformer++ | 35M | 1.3M | 14.7 | 36.23 +/- 0.11 | 1.54 +/- 0.05 |
| Primer | 34M | 1.2M | 13.8 | **35.06 +/- 0.15** | **2.13 +/- 0.11** |
| Primer-EZ | 35M | 1.1M | 13.3 | 35.16 +/- 0.13 | 2.03 +/- 0.09 |
| | *T5, TPUv2* | | | | |
| Vanilla Transformer | 35M | 2.1M | 23.9 | 23.30 +/- 0.02 | - |
| Transformer+GELU | 35M | 2.1M | 23.8 | 23.39 +/- 0.02 | 0.97 +/- 0.03 |
| Transformer++ | 35M | 2.1M | 24.2 | 23.04 +/- 0.02 | 1.33 +/- 0.05 |
| Evolved Transformer | 38M | 1.6M | 18.7 | 23.08 +/- 0.02 | 1.23 +/- 0.02 |
| Primer | 36M | 2.0M | 22.9 | 22.71 +/- 0.03 | 1.72 +/- 0.01 |
| Primer-EZ | 36M | 2.0M | 22.5 | **22.62 +/- 0.02** | **1.75 +/- 0.03** |

Table 4: Comparison on the search task, auto-regressive language modeling on LM1B, across two different hardware platforms (TPUv2s and V100 GPUs) and two different libraries (Tensor2Tensor and T5), using those libraries' default hyperparameters. This table contains the precise numbers for Figure 6. "Speedup" describes the fraction of compute used by each model to achieve the same results as the vanilla Transformer baseline trained with the full compute budget. Even though Primer was developed in Tensor2Tensor using TPUv2s, it shows strong performance on GPU and in T5. Perplexity is reported with respect to each library's default tokenization.

## A.7 Ablation and Insertion Studies

One of the core motivations of this work is to develop simple and robust Transformer modifications. To that end, we study the individual effectiveness of each Primer modification, described in Section 3 of the main text. We measure this effectiveness using insertion and ablation studies. In the insertion studies we add each modification in isolation to a vanilla Transformer. In the ablation studies we remove each modification from Primer one at a time. We are interested in how these modifications affect performance not just in our search library, Tensor2Tensor, but also in other libraries. Thus, we perform these insertion and ablation studies in a different library, T5, as a well, and use modification transferability as the key guiding metric for our modeling recommendations.

The results of these studies are shown in Figure 26. "Normalized PPLX Delta" describes the degree to which a modification helps or hurts performance. For baseline perplexity, $P_b$, and modification perplexity, $P_m$, "Normalized PPLX Delta" is defined as $\frac{P_b - P_m}{P_b}$ in the insertion study and $\frac{P_m - P_b}{P_b}$ for the ablation study. These definitions differ so that a positive value always indicates that the modification is good and a negative value always indicates that the modification is bad. Three techniques are beneficial in all scenarios. The first is "12X proj," which increases the size of the Transformer feed forward upwards projection while controlling for parameters. We find this works well for smaller models but is not useful at larger sizes. The second two, MDHA and squared ReLUs, are the defining modifications of Primer-EZ, a simpler model that captures much of the gains of the full Primer.

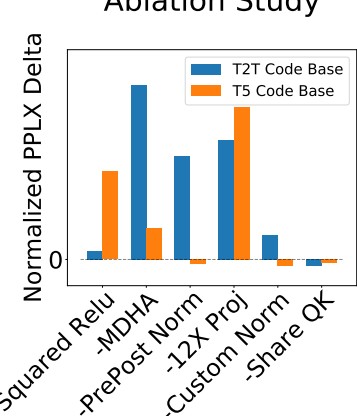

Figure 26: Investigation into transferability of Primer modifications on LM1B at ∼35M parameters. In the "Insertion Study" we insert each of the modifications into a vanilla Transformer. In the "Ablation Study," we remove each modification from Primer. "Normalized PPLX Delta" indicates the degree to which the treated models are affected by these modifications; values are normalized to be comparable across code bases and so that positive values indicate beneficial techniques in both studies. Likewise, negative values indicate harmful techniques in both studies.

## A.8    Full Training Details

In all experiments, we use previously published hyperparameter settings that were tuned for Transformer, with regularization disabled and no additional tuning for Primer. In Tensor2Tensor (T2T) these are the TRANSFORMER_TPU hyperparameters and in T5 and Lingvo these are the open-sourced parameters used in previous T5 studies [5, 49]. They both specify an Adafactor optimizer [56], with 10K warmup steps at a learning rate of 0.01, followed by reciprocal square root learning rate decay. T2T uses positional embeddings and subword tokenization, while T5 and Lingvo use relative attention [57] and SentencePieces [58].

For LM1B, we use the T2T default settings of max sequence length of 64 and batches of 4096 tokens; this is appropriate because LM1B has an average sequence length of roughly 32. For C4 and PG19, we use the T5 default of a max sequence length of 512. For one-shot pretraining, we use a max sequence length of 1024. In Section 4.2 we use batches of 65K tokens, in Section 4.3 we use batches of 1M tokens, and in Section 4.4 we uses batches of 2M tokens.

## A.9    Power Law Compute Savings Derivations

In Section 4.1 of the main text, we reproduce the results of Kaplan et al. [9] and show that, at optimal parameter sizing, the relationship between language model quality and training compute follows a power law: $l = ac^{-k}$, where $l$ is validation loss, $c$ is training compute, and $a$ and $k$ are empirical constants. This is represented as a line in double log space (Figure 7): $\log l = -k \log c + \log a$. However, these lines are not the same for each architecture we compare. The lines are roughly parallel but shifted up and down. Thus, defining the shift between two architectures' lines as $\log b^k$, we can derive the relationship of their training costs as:

$$-k \log c_0 + \log a_0 = -k \log c_1 + \log a_0 + \log b^k$$
$$-k \log c_0 = -k \log c_1 + \log b^k$$
$$c_0^{-k} = b^k c_1^{-k}$$
$$c_0 = c_1/b$$

where $b$ is a consistent reduction factor regardless of $l$. Compute savings, $s$, for using a superior architecture can now be calculated as:

$$s = c_1 - c_0$$
$$s = c_1 - c_1/b = c_1(1 - 1/b)$$
$$\text{or}$$
$$c_1 = \frac{s}{1 - 1/b}$$

Plugging this into the original power law relationship for $c_1$ we get:

$$l = a_1 \left( \frac{s}{1 - 1/b} \right)^{-k}$$
$$l = a_1(1 - 1/b)^k s^{-k}$$

Thus, the relationship between quality and compute savings yielded by an improved architecture also follows a power law with coefficient $a_1(1 - 1/b)^k$. This relationship is intuitive when recognizing that the compute reduction factor $b$ is consistent for all values of $l$ and thus a power law investment of training compute with relation to $l$ results in a power law savings with relation to $l$ as well.

### A.10 Exact T5 Numbers for Medium Sized Experiments

| Model | Params | *Baseline Compute @525K* | | | *Baseline Compute @1M* | | |
|---|---|---|---|---|---|---|---|
| | | Steps | PPLX | Speedup | Steps | PPLX | Speedup |
| | | *C4* | | | | | |
| Vanilla Transformer | 110M | 525K | 20.61 | - | 1M | 19.82 | - |
| Transformer+GELU | 110M | 524K | 20.34 | 1.20 | 998K | 19.58 | 1.26 |
| Transformer++ | 110M | 524K | 20.03 | 1.52 | 998K | 19.28 | 1.64 |
| Evolved Transformer | 110M | 351K | 20.79 | 0.89 | 668K | 19.84 | 0.98 |
| Primer | 110M | 483K | **19.82** | 1.68 | 920K | **19.07** | **1.91** |
| Primer-EZ | 110M | 471K | 19.83 | **1.71** | 896K | **19.07** | 1.90 |
| Switch Transformer | 550M | 525K | 17.16 | - | 1M | 16.32 | - |
| Switch Primer | 550M | 474K | **16.56** | **1.45** | 900K | **15.82** | **1.56** |
| Synthesizer | 145M | 525K | 20.35 | - | 1M | 19.57 | - |
| + Squared ReLU | 145M | 523K | **19.55** | **1.74** | 996K | **18.83** | **1.96** |
| | | *PG19* | | | | | |
| Vanilla Transformer | 110M | 525K | 16.39 | - | 1M | 15.83 | - |
| Transformer+GELU | 110M | 524K | 16.35 | 1.01 | 998K | 15.84 | 0.95 |
| Transformer++ | 110M | 524K | 16.15 | 1.18 | 998K | 15.64 | 1.20 |
| Primer | 110M | 483K | 15.96 | 1.68 | 920K | **15.31** | 1.81 |
| Primer-EZ | 110M | 471K | **15.84** | **1.74** | 896K | 15.37 | **1.98** |

Table 5: Language modeling comparison on larger datasets, transferring Primer to the T5 codebase. In this transferred regime, Primer improves upon all baselines. Furthermore, Primer-EZ not only reaches parity with Primer, but in some cases, surpasses it. Switch Transformer and Synthesizer also benefit from the Primer-EZ modifications. Compute budget comparison points are chosen according to how long it takes vanilla baselines to reach 525K and 1M training steps. Perplexities are given with respect to SentencePieces. This table has the precise numbers for Figure 9.

## A.11 Performance on Individual One-Shot Tasks

| Task | Metric | Transf. 1/3 | Transf. Full | Primer 1/3 | Primer Full | GPT-3 XL |
|---|---|---|---|---|---|---|
| Pretraining | pplx | 15.3 | 14.3 | 14.3 | 13.5 | - |
| *Question Answering Tasks* | | | | | | |
| TriviaQA | acc | $22.5 \pm 0.4$ | $26.8 \pm 0.5$ | $27.5 \pm 0.4$ | $\mathbf{32.2 \pm 0.5}$ | 26.5 |
| WebQs | acc | $9.1 \pm 0.5$ | $9.6 \pm 0.4$ | $9.8 \pm 0.8$ | $10.4 \pm 0.3$ | 9.2 |
| NQs | acc | $5.8 \pm 0.2$ | $6.7 \pm 0.2$ | $\mathbf{7.8 \pm 0.5}$ | $\mathbf{9.1 \pm 0.3}$ | 5.4 |
| SQuADv2 | f1 | $54.2 \pm 2.4$ | $65.4 \pm 2.9$ | $64.2 \pm 3.7$ | $67.8 \pm 1.2$ | 54 |
| CoQa | f1 | $52.5 \pm 1.1$ | $57.7 \pm 1.2$ | $59.1 \pm 0.9$ | $\mathbf{61.2 \pm 0.7}$ | 66.1 |
| DROP | f1 | $21.5 \pm 0.4$ | $23.4 \pm 0.2$ | $\mathbf{24.8 \pm 0.5}$ | $\mathbf{26.5 \pm 0.2}$ | 23 |
| Quac | f1 | $30.1 \pm 0.5$ | $30.9 \pm 0.7$ | $28.9 \pm 0.9$ | $30.2 \pm 0.7$ | 32.3 |
| LAMBADA | acc | $51.5 \pm 0.9$ | $55.2 \pm 1.3$ | $54.5 \pm 1.1$ | $56.8 \pm 0.9$ | 58.3 |
| QA Average | avg | 30.9 | 34.5 | 34.6 | 36.8 | 34.3 |
| *Multi-Choice Schema Tasks* | | | | | | |
| HellaSwag | acc | $55.7 \pm 0.3$ | $59.5 \pm 0.2$ | $\mathbf{60.2 \pm 0.3}$ | $\mathbf{63.3 \pm 0.2}$ | 53.5 |
| StoryCloze | acc | $75.2 \pm 0.3$ | $75.9 \pm 0.4$ | $\mathbf{76.9 \pm 0.2}$ | $\mathbf{77.5 \pm 0.3}$ | 74.2 |
| Winogrande | acc | $55.4 \pm 0.3$ | $58.4 \pm 0.4$ | $58.8 \pm 0.3$ | $\mathbf{60.4 \pm 0.2}$ | 59.1 |
| PIQA | acc | $72.6 \pm 0.5$ | $72.6 \pm 0.3$ | $73.7 \pm 0.5$ | $\mathbf{75.0 \pm 0.4}$ | 74.4 |
| ARC (Challenge) | acc | $32.7 \pm 0.4$ | $34.4 \pm 0.3$ | $35.6 \pm 0.9$ | $\mathbf{37.4 \pm 0.4}$ | 36.4 |
| ARC (Easy) | acc | $64.5 \pm 0.5$ | $64.9 \pm 0.5$ | $65.6 \pm 0.6$ | $\mathbf{67.5 \pm 0.5}$ | 55.9 |
| OpenBookQA | acc | $45.3 \pm 0.9$ | $46.8 \pm 0.8$ | $47.9 \pm 0.4$ | $\mathbf{49.3 \pm 0.5}$ | 46.4 |
| ANLI R1 | acc | $33.9 \pm 1.2$ | $35.5 \pm 0.2$ | $35.5 \pm 0.4$ | $34.8 \pm 0.3$ | 34.6 |
| ANLI R2 | acc | $33.5 \pm 0.7$ | $33.4 \pm 0.5$ | $34.5 \pm 0.6$ | $33.5 \pm 0.4$ | 32.7 |
| ANLI R3 | acc | $34.5 \pm 0.7$ | $35.2 \pm 0.1$ | $33.0 \pm 0.3$ | $33.8 \pm 0.5$ | 33.9 |
| ReCoRD | acc | $84.8 \pm 0.1$ | $86.3 \pm 0.2$ | $85.8 \pm 0.3$ | $\mathbf{86.7 \pm 0.0}$ | 83 |
| WSC | acc | $67.4 \pm 0.8$ | $66.8 \pm 1.2$ | $69.3 \pm 1.3$ | $68.9 \pm 1.2$ | 62.5 |
| BoolQ | acc | $58.9 \pm 1.1$ | $63.6 \pm 2.1$ | $60.7 \pm 0.8$ | $64.7 \pm 2.0$ | 63.7 |
| CB | acc | $56.3 \pm 2.5$ | $53.0 \pm 2.7$ | $55.4 \pm 3.3$ | $56.6 \pm 9.6$ | 48.2 |
| RTE | acc | $48.4 \pm 1.2$ | $53.6 \pm 2.5$ | $54.3 \pm 1.5$ | $52.9 \pm 2.8$ | 49.5 |
| COPA | acc | $80.2 \pm 3.2$ | $87.2 \pm 1.2$ | $84.8 \pm 1.5$ | $87.5 \pm 1.1$ | 74 |
| WiC | acc | $51.6 \pm 0.2$ | $51.0 \pm 0.5$ | $\mathbf{51.7 \pm 0.1}$ | $\mathbf{51.8 \pm 0.1}$ | 49.2 |
| RACE-h | acc | $39.4 \pm 0.4$ | $40.8 \pm 0.4$ | $40.4 \pm 0.4$ | $\mathbf{43.7 \pm 0.3}$ | 42 |
| RACE-m | acc | $50.0 \pm 1.0$ | $52.6 \pm 0.4$ | $51.8 \pm 0.8$ | $\mathbf{54.0 \pm 0.4}$ | 55.2 |
| Multi-Choice Average | avg | 53.1 | 54.7 | 55 | 56.2 | 54.1 |

Table 6: Comparison between Transformer+GELU and Primer at 1.9B parameters on downstream one-shot tasks at 1/3 and full pretraining compute budgets. One-shot sample means and standard deviations are computed using the evaluated performance of 5 weight checkpoints. **Bold numbers** denote improved one-shot performance and shaded numbers denote worse one-shot performance compared to Transformer with full compute that is statistically significant under an independent t-test with p-value threshold 0.05. Primer achieves the same performance as Transformer when given 1/3 the training compute and stronger performance on a majority of tasks when given the same training compute. GPT-3 XL [7] scores are provided as a grounding reference point; they should not be closely compared to our results as the models have different pretraining configurations.

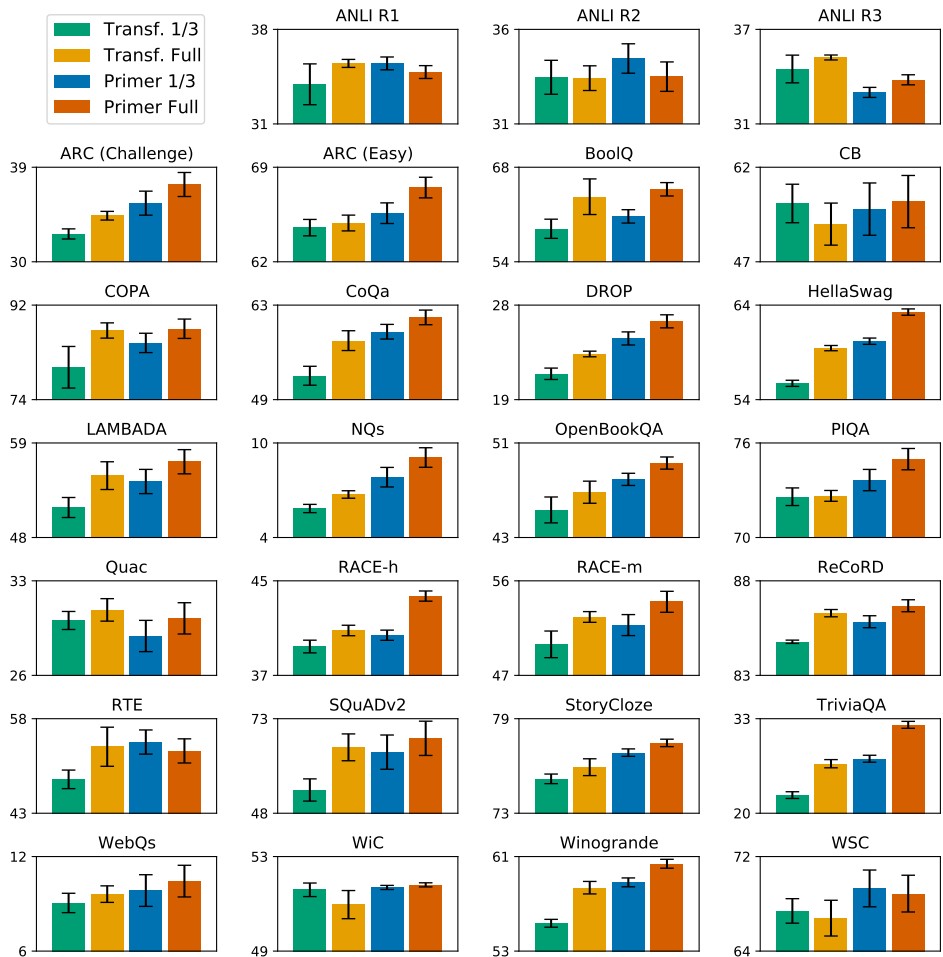

Figure 27: Comparison between Transformer+GELU and Primer at 1.9B parameters on downstream one-shot tasks at 1/3 and full pretraining compute budgets. 95% confidence intervals are provided according to an independent t-test, using a sample of 5 pretraining weight checkpoints. Primer achieves roughly the same performance as Transformer when given 1/3 the pretraining compute and stronger performance on a majority of tasks when given the same pretraining compute. Exact numbers are presented in Table 6.

## A.12   Masked Language Modeling

Encoder-decoder style masked language modeling (MLM) is not the focus of this work. However, because it was the focus of the original T5 project, we include MLM comparisons here for completeness (Table 7). Specifically, we use the exact comparison configuration used by Narang et al.[49], who benchmarked several Transformer variants; the one difference is that we only run model training one time, since this regime is not the focus of our study. For "Primer-EZ Decoder" we use a Transformer++ encoder and a Primer-EZ decoder. Our treatments demonstrate that the Primer-EZ modifications have the capacity to improve encoder-decoder MLM models, but perhaps to a lesser degree, when compared to Transformer++. We believe this indicates that decoder-only LM and encoder-decoder MLM benefit from different modeling decisions – something that could be studied in future works. We also believe that running our search on encoder-decoder MLM directly could yield modifications that are more beneficial for this task.

| Model | Params | Pretraining Log PPLX | SGLUE | XSum | WebQ |
|-------|--------|---------------------|-------|------|------|
| Vanilla Transformer* | 223M | 1.838 | 70.97 | 17.78 | 23.02 |
| Transformer+GeLU* | 223M | 1.838 | 73.67 | 17.86 | 25.13 |
| Transformer++ | 224M | 1.792 | 75.65 | **17.90** | **25.92** |
| Primer-EZ Decoder | 224M | **1.787** | **76.69** | 17.87 | 24.87 |

Table 7: Masked language modeling comparison on C4 in T5 with encoder-decoder style models. These results are run in the exact same configuration as Narang et al. [49], although we only run our models once, as MLM is not the focus of our work. * indicates rows that are taken from that study.

### A.13 Carbon Emission Estimates

Following the recommendations of Patterson et al. [59], we release the carbon emission estimates for our largest experiments.

To estimate the carbon emissions[3,4] for our architecture search, we build off of the measurements taken by Patterson et al. Their emissions estimate for architecture search is 3.2 MTCO$_2$e for 1360 days of TPUv2 usage [59]. Here, we use 1145.8 days of TPUv2 compute for our search. Additionally, the PUE for our data center[5] at the time of our search was 1.08 instead of 1.10, and its net carbon intensity average was 0.336 MTCO$_2$e/MWh instead of 0.431 MTCO$_2$e/MWh.[6,7] Thus, the proportional emissions estimate for our architecture search experiments is 3.2 MTCO$_2$e $*\frac{1145.8}{1360} * \frac{1.08}{1.10} * \frac{336}{431} = 2.06$ MTCO$_2$e. For comparison, a round trip plane ticket from San Francisco to New York for a single passenger is $\sim$1.2 MTCO$_2$e [59] and so our search costs roughly 1.72 such plane tickets.

We follow the same process of building off of the Patterson et al. measurements to estimate emissions for our large scale T5 experiments. The Patterson et al. emissions estimate for 11B parameter T5 is 46.7 tCO$_2$e for 10,249 days of TPUv3 usage. Our T5 models are smaller, and so only require 687.5 TPUv3 days to train on average. We run 3 trainings (Primer, original T5 and T5++) to show Primer's improvements over baselines, yielding a total of 2062.5 TPUv3 days. When we ran our experiments, the data center[8] PUE was 1.10 instead of 1.12 and its net carbon intensity average was 0.540 MTCO$_2$e/MWh instead of 0.545 MTCO$_2$e/MWh. Thus, the proportional total estimate for these T5 model trainings is 46.7 MTCO$_2$e $*\frac{2062.5}{10,249} * \frac{1.10}{1.12} * \frac{540}{545} = 8.54$ MTCO$_2$e.

To estimate the emissions of our one-shot pretrainings in Lingvo, we measure system average power in the same manner as Patterson et al. [59]. Including memory, network interface, fans, and host CPU, the average power per TPUv4 chip is 343W. We use the same equation as Patterson et al. to calculate

---

[3]Our CO$_2$e accounting methodology for data center net carbon intensity does not currently fit the Greenhouse Gas (GHG) protocol for emissions reporting (Scope 2 and 3 for electricity). This deviation is due to a change in methodology where Google uses hourly life cycle emission factors, while the GHG Protocol generally relies on annual operating emission factor data. Google chooses to share these modified metrics as part of our 24/7 carbon-free energy (CFE) program, focused on our goal of achieving 100% 24/7 local CFE by 2030. Google's target for 2030 goes beyond the traditional Scope 2 rules to restrict both the location and the accounting period. This means that, instead of anywhere in a continent, the CFE purchase should be on the same geographically local grid; and instead of the accounting period being one year, the accounting should be within the same hour.

[4]While electricity consumption is relatively straightforward, strategies to reduce greenhouse gas emissions are not. For details on the distinction between conventional carbon offsets, Google's goal for 2030 of 24/7 CFE for its global data centers and campuses, and what it is doing now to set the groundwork for 2030, please see Appendix B of Patterson et al. [59].

[5]Each data center is located within a Regional Grid, which is the geographic basis for Google's 24/7 CFE goals. For our data center in Georgia, the Regional Grid is the Southern Company balancing authority.

[6]The net carbon intensity at a particular data center is based on accounting for hourly emission reductions via real time, local carbon-free energy purchases. This is calculated using the 24/7 carbon-free energy methodology, which can be reviewed in greater depth in "24/7 Carbon-Free Energy: Methodologies and Metrics" [60].

[7]The carbon intensity values utilized in this paper are at the annual 2020 grid level for each data center in which the models were run.

[8]For our data center in Taipei, for purposes of Google's 24/7 CFE accounting, the Regional Grid is Taiwan.

$CO_2$e for our 2 large scale pretrainings: $2 * 343W * 71,800h * 1.08(PUE) * 0.055$ MTCO$_2$e/MWh $=$ 29.26 MTCO$_2$e.[9]

The emission cost for our large scale T5 and one-shot comparisons are higher than the cost of the architecture search itself. We invest in these large scale comparisons to demonstrate the potential savings of our efficient modifications. For instance, the savings for using Primer over Transformer described in Section 4.4 of the main text equates to 9.75 MTCO$_2$e, which alone is ∼4.7X the cost of the architecture search. Note, differences in hardware setups affect these savings. For example, the one-shot models were trained in Oklahoma, which has favorable MTCO$_2$e/MWh when compared to Georgia, where the Primer search was conducted. To factor out the effects of these hardware differences, we can instead analyze Primer's return on investment in terms of FLOPs. The search for Primer cost ∼2.14E+21 FLOPs. Training Transformer for the one-shot comparison cost ∼2.96E+22 FLOPs, which means the compute saved by Primer is ∼1.98E+22 FLOPs, given that it only requires a third of the compute to achieve the same quality. Thus, Primer's savings in the one-shot experiments are 9.24X the cost of the architecture search itself, yielding returns on investing in the search. Note that the search cost is a one-time cost and that Primer can be reused in future trainings to save more compute. For example, our largest models are roughly 100X smaller than the full scale GPT-3 [7], and so the return on our search investment can grow if Primer is scaled up to larger training configurations.

## A.14 Comparison to Evolved Transformer

| Model | LM1B | | C4 | |
| --- | --- | --- | --- | --- |
| | Params | PPLX @ 1.5M Steps | Params | PPLX @ 1M Steps |
| Vanilla Transformer | 35M | 23.45 | 110M | 19.82 |
| Transformer+GELU | 35M | 23.68 | 110M | 19.58 |
| Transformer++ | 35M | 23.35 | 110M | 19.29 |
| Evolved Transformer | 38M | 23.11 | 110M | 19.37 |
| Primer | 36M | 22.97 | 110M | 18.99 |
| Primer-EZ | 36M | **22.89** | 110M | **18.93** |

Table 8: Auto-regressive language modeling comparison between Primer and various baselines, including the Evolved Transformer, controlling for training steps in T5. These are the same experiments featured in Tables 4 and 5, but with the data presented to compare sample efficiency instead of training compute efficiency.

This work builds off of the Evolved Transformer [13], which also sought to discover improved sequence models using architecture search. Compute efficiency comparisons to the Evolved Transformer architecture are provided in T5 on LM1B in Table 4 and on C4 in Table 5. Sample efficiency comparisons to the Evolved Transformer architecture are offered in Table 8 on those same experiments. In this section we discuss these comparisons and how they highlight the improvements of our Primer search over the Evolved Transformer search.

Firstly, our Primer search aims to improve training compute efficiency, which yields more practical results than the sample efficiency objective of So et al. [13], who controlled for number of train steps when evaluating models. Evolved Transformer is effective in this controlled-train-step regime when comparing to other baselines, as shown in Table 8. When controlling for number of training steps in this way, Evolved Transformer is roughly on par with Transformer++ on C4 and is better than Transformer++ on LM1B. However, Evolved Transformer is substantially slower than all other models (see Tables 4 and 5) because it is deeper; we follow the same scaling policy as So et al. of adding additional layers to control for parameters, given that an Evolved Transformer layer has significantly less parameters than a standard Transformer layer. Evolved Transformer's slowness counteracts its sample efficiency and for this reason its speedup factor is diminished on LM1B and less than 1.0 (indicating a slowdown over vanilla Transformer) on C4 (see Tables 4 and 5). This limits Evolved Transformer's practicality. In contrast, Primer is designed to specifically address this shortcoming and thus delivers the practical result of substantial compute savings.

---

[9]For our data center in Oklahoma, for purposes of Google's 24/7 CFE accounting, the Regional Grid is the Southwest Power Pool (SPP) Independent System Operator.

The open-ended nature of the Primer search also allows for effective modifications that were not available to the Evolved Transformer search. In fact, *none* of the Primer modifications (see Section 3) can be represented in the Evolved Transformer search space, aside from resizing hidden dimension sizes. This is because the Evolved Transformer search space followed a rigid ordering of components and used a vocabulary of unalterable high level building blocks. For example, normalization always preceded weighted transformations and, although there were different weighted transformations to choose from such as self-attention and GLU, those transformations could not be modified by the search. In contrast, the Primer search space allows for the modification of all initialized modules – such as weighted transformations, activation functions and normalization functions – as well as allows for macro-level reordering, such as moving normalization after weighted transformations. We believe that this difference in openness is what allowed Primer to develop definitively superior modifications, as demonstrated not only by improved compute efficiency, but also by improved sample efficiency (Table 8), which is what Evolved Transformer was meant to optimize.

### A.15 Practical Discussion

The main motivation of this work is to develop simple and practical changes to Transformers that can be easily adopted. To that end, we provide answers to some questions that practitioners may ask:

- *Are the Primer training compute savings going to be the same in all setups?* No. Across our own provided experiments, Primer yields various compute savings. This is because the compute savings depend on hardware specifics, deep learning library operation speeds, model sample efficiencies on specific tasks, and other factors that may vary across setups. We use the exact replica of T5 training as a demonstration of what savings look like in an established configuration (4.2X), but expect results to vary across configurations.

- *Can Primer improve BERT [2]?* This work has focused on the specific task of auto-regressive language modeling, which, with the development of GPT-3, proves to be important for both traditional NLP applications as well as generative applications. We have only briefly investigated Primer's application to masked language modeling and encoder-decoder models (Appendix A.12). Our investigations show that, while Primer improves upon vanilla Transformer, it is not obviously better than Transformer++. Thus, modifications that work well for auto-regressive language modeling may not be as effective for masked language modeling. Future work could investigate if the Primer modifications can be integrated into encoder-decoder and encoder-only models in a more effective way that can improve models like BERT. Future work could also apply the search method described here to finding better encoder-based masked language models.

- *Do hyperparameter configurations need to be retuned to use Primer?* Our intention is for Primer modifications to not require any additional hyperparameter tuning. To that end, in our experiments we did not tune any hyperparameters, and instead used the Transformer hyperparameters from established libraries. However, Primer may work even better with additional tuning.

- *Is Primer-EZ better than Primer?* In our comparison experiments, we find that Primer-EZ is sometimes better than Primer in the T5 codebase. However, in application to other codebases, such as Lingvo and T2T, we find that the full Primer can give improved performance over Primer-EZ. Thus, we recommend that practitioners first try using Primer-EZ for its ease of implementation and then move on to implementing the full Primer if they are interested in achieving further gains.