# OpenReview forum: "Searching for Efficient Transformers for Language Modeling"
_NeurIPS.cc/2021/Conference — NeurIPS 2021 Poster_

### Official Review · Reviewer_WYyB · 2021-07-12

**Rating:** 7
**Confidence:** 4

**Summary:**

This paper studies how to improve the transformer architecture for the task of _autoregressive_ language modeling. To do so, they start from the transformer architecture and do low-level architecture search. They find an architecture, Primer, which provides better training efficiency / scaling laws over the original transformer. Most interestingly, they show that a simple variant of the transformer Primer-EZ, which simply adds a convolution layer and a squaring operation to the ReLU activations, can provide similar or better improvements than the full Primer. They test their implementation of Primer-EZ in multiple codebases and datasets, and show that the gains transfer.

**Limitations And Societal Impact:**

Yes, this work can improve training efficiency and reduce costs across the field.

**Main Review:**

Strengths:
* I really like the idea of Primer-EZ, where they strip down all the complexity induced by the architecture search and focus on the smallest number of modifications that they can make while still getting good results. One of the downsides of typical NAS is that the resulting architecture is a mess and impossible to implement from scratch---this is fortunately not true for Primer-EZ.
* I like that they tested across numerous codebases and datasets. My takeaway from this part was that Primer-EZ actually works, as opposed to other transformer modifications which arent reproducible across different codebases https://arxiv.org/abs/2102.11972.

Weaknesses:
* The only thing that would make this a “home run” paper for me would be experiments that show gains on masked language model pretraining + finetuning. Right now it seems that Transformer++ is a strong baseline in this setting (see the Appendix). However, I agree with the authors that this is out of scope and that focusing on autoregressive language modeling itself is a fine goal.
* I would like to see more discussion and comparison with the Evolved Transformer architecture https://arxiv.org/abs/1901.11117.

**Time Spent Reviewing:**

2

---

> ### Author Response · Authors · 2021-08-10
> **Response to Reviewer WYyB**
>
> Thank you for taking the time to review our work and provide feedback. We respond to quoted comments below:
>
> \
> \
> *“The only thing that would make this a ‘home run’ paper for me would be experiments that show gains on masked language model pretraining + finetuning. Right now it seems that Transformer++ is a strong baseline in this setting (see the Appendix). However, I agree with the authors that this is out of scope and that focusing on autoregressive language modeling itself is a fine goal.”*
>
> We intend on releasing additional results that demonstrate Primer’s gains transfer to downstream oneshot tasks. Specifically, we train Primer in a training configuration similar to GPT-3 at 1B parameters and demonstrate that its pretraining perplexity improvement over Transformer does indeed translate to downstream one-shot improvement.
>
> As far as masked language modeling specifically is concerned, we acknowledge that applying our changes to encoder models is not as effective and that that is a limitation of our work.
>
> \
> \
> *“I would like to see more discussion and comparison with the Evolved Transformer architecture https://arxiv.org/abs/1901.11117.”*
>
> \
> We train Evolved Transformer in the T5 LM1B setting (comparable to results in Figure 6 and Table 4):
>
> |        Model        | Params | Train Steps Step/Sec  | PPLX @ 24 Hours |    Speedup    | PPLX @1.5M Steps |
> |:-------------------:|:------:|:---------------------:|:---------------:|:-------------:|:----------------:|
> | Vanilla Transformer |   35M  |          23.9         |  23.30 +/- 0.02 |       -       |  23.45 +/- 0.25  |
> | Transformer+Gelu    |   35M  |          23.8         |  23.39 +/- 0.02 | 0.97 +/- 0.03 |  23.68 +/- 0.03  |
> | Transformer++       |   35M  |          24.2         |  23.04 +/- 0.02 | 1.55 +/- 0.31 |  23.35 +/- 0.02  |
> | Evolved Transformer |   38M  |          18.7         |  23.08 +/- 0.02 | 1.23 +/- 0.02 |  23.11 +/- 0.01  |
> | Primer              |   37M  |          22.9         |  22.71 +/- 0.03 | 1.72 +/- 0.01 |  22.97 +/- 0.03  |
> | Primer-EZ           |   36M  |          22.9         |  22.64 +/- 0.02 |  1.8 +/- 0.03 |  22.89 +/- 0.03  |
>
> \
> We also train Evolved Transformer in the T5 C4 setting (comparable to results in Figure 9 and Table 5):
>
> |        Model        | Params | Steps @ 242 Hours | PPLX @ 242 Hours | Speedup | PPLX @ 525K Steps |
> |:-------------------:|:------:|:-----------------:|:----------------:|:-------:|:-----------------:|
> | Vanilla Transformer |  110M  |         1M        |       19.82      |    -    |       20.61       |
> | Transformer+Gelu    |  110M  |        998K       |       19.58      |   1.26  |       20.33       |
> | Transformer++       |  110M  |        998K       |       19.28      |   1.64  |       20.02       |
> | Evolved Transformer |  110M  |        668K       |       19.84      |   0.98  |       20.16       |
> | Primer              |  110M  |        920K       |       19.07      |   1.81  |       19.72       |
> | Primer-EZ           |  110M  |        909K       |       19.02      |   1.98  |       19.65       |
>
> \
> We first note some performance observations. On both tasks, Evolved Transformer is significantly slower per training step than all other models. This is because Evolved Transformer is deeper than all other models -- we follow the same scaling policy as So et al. to equalize the number of parameters, considering an Evolved Transformer layer has significantly less parameters than a standard Transformer layer. In terms of perplexity at a given train step, Evolved Transformer is competitive with the other baselines in both cases, beating Transformer++ on LM1B. However, Evolved Transformer’s long train step time counteracts its sample efficiency and for this reason its speedup factor is diminished on LM1B and less than 1.0 (indicating a slowdown over vanilla Transformer) on C4.
>
> These deficits highlight the benefits of the Primer search over the Evolved Transformer search. Firstly, the Evolved Transformer search looked for the highest accuracy architecture at a fixed number of train steps. The Primer search targets the highest accuracy architecture given a fixed compute budget. This explains why Primer is the more practical option, more drastically lowering training costs. Secondly, even when considering only sample efficiency, Primer outperforms the Evolved Transformer in both cases. We attribute this to Primer’s modifications being more effective at improving accuracy -- a result of the Primer search space’s ability to make low-level modifications, such as mutating activation functions and self attention mechanisms; this contrasts the Evolved Transformer’s search space, composed only of high level building blocks that could not be altered. For example, the Evolved Transformer decoder utilizes depthwise convolution via. depthwise separable convolution feed forward layers, but Primer’s simple addition of depthwise convolution within the attention mechanism itself has proved to be more beneficial.
>
> \
> \
> *“I really like the idea of Primer-EZ, where they strip down all the complexity induced by the architecture search and focus on the smallest number of modifications that they can make while still getting good results. One of the downsides of typical NAS is that the resulting architecture is a mess and impossible to implement from scratch---this is fortunately not true for Primer-EZ.”*
>
> Thank you for recognizing this! These were our exact thoughts when creating Primer-EZ.

---

> ### Author Response · Authors · 2021-08-27
> **Second Response to Reviewer WYyB (One-shot Results)**
>
> As noted in our first response, we have conducted additional experiments that mimic the one-shot GPT-3 setup (https://arxiv.org/pdf/2005.14165.pdf). GPT-3 was not open sourced, and so this is not an exact replication, but we have done our best to reproduce the results using a proprietary pretraining dataset and the same one-shot downstream tasks. The results of a 1.9B parameter comparison between Primer and Transformer+GELU (approximating the GPT-3 architecture) in Lingvo are presented below. Note, these are *not* directly comparable to GPT-3, as our configuration is different, but we provide the GPT-3 numbers as well to demonstrate that our results are reasonable for the given model sizes. We train Transformer (at full compute) for 1M steps so that it is roughly on par with the performance of GPT-3 XL.
>
> \
> First, we present crude aggregates that summarize the high level trend that Primer can achieve the same results as Transformer+GELU using ⅓ of the pretraining compute and better results when given the same pretraining compute:
>
> |                             | Transformer 1/3 Compute | Transformer Full Compute | Primer 1/3 Compute | Primer Full Compute | GPT-3 XL (Brown et al., 2020) |
> |-----------------------------|:-----------------------:|:------------------------:|:------------------:|:-------------------:|:-----------------------------:|
> |     Pretraining Val PPLX    |           15.3          |           14.3           |        14.3        |         13.5        |               -               |
> |    Mean QA One-Shot Score   |           30.9          |           34.5           |        34.6        |         36.8        |              34.3             |
> | Mean Multi-Choice One-Shot Score |           53.1          |           54.7           |         55         |         56.2        |              54.1             |
>
> \
> \
> We now present the exact scores for each task:
>
> |                 | Transformer 1/3 Compute | Transformer Full Compute | Primer 1/3 Compute | Primer Full Compute | GPT-3 XL (Brown et al., 2020) |
> |-----------------|:-----------------------:|:------------------------:|:------------------:|:-------------------:|:-----------------------------:|
> |     TriviaQA    |    % 22.49 +/- 0.37 %   |      26.81 +/- 0.45      |   27.50 +/- 0.38   | # 32.16 +/- 0.47 #|              26.5             |
> |      WebQs      |      9.06 +/- 0.50      |       9.62 +/- 0.42      |    9.84 +/- 0.81   |    10.44 +/- 0.34   |              9.15             |
> |       NQs       |    % 5.84 +/- 0.21 %    |       6.73 +/- 0.19      |  # 7.83 +/- 0.50 # |  # 9.08 +/- 0.30 #  |              5.43             |
> |     SQuADv2     |    % 54.15 +/- 2.36 %   |      65.43 +/- 2.86      |   64.15 +/- 3.65   |    67.81 +/- 1.15   |               54              |
> |     LAMBADA     |    % 51.49 +/- 0.94 %   |      55.20 +/- 1.30      |   54.52 +/- 1.14   |    56.81 +/- 0.94   |              58.3             |
> |       CoQa      |    % 52.54 +/- 1.13 %   |      57.74 +/- 1.18      |   59.06 +/- 0.87   |  # 61.18 +/- 0.65 # |              66.1             |
> |       DROP      |    % 21.46 +/- 0.44 %   |      23.35 +/- 0.22      | # 24.83 +/- 0.50 # |  # 26.46 +/- 0.24 # |               23              |
> |       Quac      |      30.06 +/- 0.54     |      30.85 +/- 0.67      |   28.91 +/- 0.93   |    30.22 +/- 0.65   |              32.3             |
> |                 |                         |                          |                    |                     |                               |
> |    HellaSwag    |    % 55.73 +/- 0.26 %   |      59.45 +/- 0.22      | # 60.18 +/- 0.27 # |  # 63.26 +/- 0.20 # |              53.5             |
> |    StoryCloze   |      75.18 +/- 0.25     |      75.92 +/- 0.43      | # 76.85 +/- 0.19 # |  # 77.46 +/- 0.34 # |              74.2             |
> |    Winogrande   |    % 55.35 +/- 0.27 %   |      58.37 +/- 0.43      |   58.82 +/- 0.30   |  # 60.39 +/- 0.22 # |              59.1             |
> |       PIQA      |      72.59 +/- 0.45     |      72.64 +/- 0.28      |   73.65 +/- 0.54   |  # 74.97 +/- 0.35 # |              74.4             |
> | ARC (Challenge) |    % 32.65 +/- 0.39 %   |      34.39 +/- 0.33      |   35.58 +/- 0.92   |  # 37.36 +/- 0.36 # |              36.4             |
> |    ARC (Easy)   |      64.53 +/- 0.49     |      64.87 +/- 0.47      |   65.60 +/- 0.62   |  # 67.49 +/- 0.53 # |              55.9             |
> |     ANLI R1     |      33.93 +/- 1.22     |      35.48 +/- 0.24      |   35.48 +/- 0.38   | % 34.84 +/- 0.25 %  |              34.6             |
> |     ANLI R2     |      33.47 +/- 0.72     |      33.42 +/- 0.53      |   34.46 +/- 0.63   |    33.50 +/- 0.37   |              32.7             |
> |     ANLI R3     |      34.49 +/- 0.71     |      35.22 +/- 0.13      | % 33.00 +/- 0.26 % |  % 33.80 +/- 0.47 % |              33.9             |
> |      ReCoRD     |    % 84.78 +/- 0.07 %   |      86.29 +/- 0.16      |   85.84 +/- 0.26   |  # 86.68 +/- 0.04 # |               83              |
> |       WSC       |      67.40 +/- 0.84     |      66.79 +/- 1.22      |   69.30 +/- 1.25   |    68.86 +/- 1.23   |              62.5             |
> |      BoolQ      |    % 58.89 +/- 1.14 %   |      63.63 +/- 2.12      |   60.72 +/- 0.80   |    64.73 +/- 1.98   |              63.7             |
> |        CB       |      56.25 +/- 2.46     |      52.98 +/- 2.69      |   55.36 +/- 3.34   |    56.55 +/- 9.63   |              48.2             |
> |       RTE       |    % 48.38 +/- 1.19 %   |      53.55 +/- 2.50      |   54.30 +/- 1.54   |    52.89 +/- 2.81   |              49.5             |
> |       COPA      |    % 80.17 +/- 3.19 %   |      87.17 +/- 1.17      |   84.80 +/- 1.48   |    87.50 +/- 1.05   |               74              |
> |       WiC       |      51.59 +/- 0.23     |      50.97 +/- 0.48      | # 51.69 +/- 0.07 # |  # 51.80 +/- 0.09 # |              49.2             |
> |    OpenBookQA   |      45.33 +/- 0.88     |      46.83 +/- 0.75      |   47.92 +/- 0.41   |  # 49.30 +/- 0.45 # |              46.4             |
> |      RACE-h     |    % 39.44 +/- 0.43 %   |      40.80 +/- 0.36      |   40.40 +/- 0.35   |  # 43.71 +/- 0.33 # |               42              |
> |      RACE-m     |    % 49.95 +/- 1.02 %   |      52.56 +/- 0.41      |   51.79 +/- 0.80   |  # 54.00 +/- 0.42 # |              55.2             |
>
> Sample means and standard deviations are computed using the evaluated performance of 5 weight checkpoints.
>
> \# denotes improved performance over Transformer with full compute that is statistically significant under independent two-sample t-test with p-value threshold 0.05.
>
> % denotes worse performance than Transformer with full compute that is statistically significant under independent two-sample t-test with p-value threshold 0.05.
>
> \
> Primer with ⅓ compute achieves equal performance with Transformer using full compute on 21 tasks, better performance on 5 tasks and worse performance on 1 task. Primer with full compute achieves equal performance with Transformer using full compute on 10 tasks, better performance on 15 tasks and worse performance on 2 tasks.
>
> Thus, we believe the benefits of Primer extend beyond pure language modeling, and can be applied to the pretraining/one-shot regime. This can increase Primer's community impact, especially as large language model training continues to grow in size with the increasing popularity of pretraining.

---

### Official Review · Reviewer_dssT · 2021-07-17

**Rating:** 5
**Confidence:** 4

**Summary:**

This paper mainly investigates how to perform architecture search to obtain the efficient transformer for auto-regressive language modeling. Compared with previous works, this searched model introduces two simple and effective modifications: squaring ReLU and depthwise convolution. Experimental results also demonstrate that the proposed model can achieve comparable results with fewer computations.

**Limitations And Societal Impact:**

Yes

**Main Review:**

Strengths:

1. This paper applies neural architecture search, which adopts regularized evolution and conceptual initialization, to obtain the efficient transformer model for language modeling tasks.
2. Experimental results have validated the effectiveness and efficiency of the searched model.

Weaknesses:

1. The contribution of this paper is incremental. First, the searching algorithm is based on the existing works (Evolutionary search). The main difference is this paper aims to obtain a trade-off between the training time and sample efficiency, but what is the optimal trade-off has not been discussed. Second, as a NAS method, the searching algorithm is more important than the searched model. However, the authors cost many columns for introducing the searched model.
2. The part of the introduction section is like a program instruction. I am not sure whether it is appropriate for a research paper.
3. Authors highlight that the multi-DConv-Head attention is important, but no ablation study has been conducted for analyzing the impact of this module.

Questions:

1. Is Tensorflow Program necessary? If the core contributions of this paper are the searching algorithm and the searched models, I think it should not be sensitive to the code platform (like PyTorch or MXNet). In other words, I think it is unnecessary to highlight Tensorflow usage in a research paper.
2. This paper argues the improvements are mainly from Squared ReLU and Multi-DConv-Head Attention. So these modules are determined by neural architecture search, or just hand-crafted design?

Comments:

1. Line 115, "search search" => "search"

=======================

After read authors' response, I have raised my score to 5. But I still lean negative opinion on this paper. Here are my responses:
1.	Just as aforementioned, your paper focuses more attention on the model (named Primer) rather than the searching algorithm. Therefore, I think your title needs to be changed, like "Primer: xxx" rather than "searching for efficient transformer". Otherwise, it is unfair for other NAS papers, let alone you have not compared with any other NAS-based methods. And your paper is mainly focused on the autoregressive language model, called "for language modeling", which is also a little over-claim.
2.	Searching efficient transformer has been surveyed in many previous works [a][b][c]. The difference between these works is that they only focus on lightweight compressed transformers due to the limitation of resources.  Considering no one has searched such a large-scale model previously, it can be viewed as a contribution and I have raised my score to 5 based on this merit. However, a question is if under the same computation, is it possible to obtain a better model than other NAS methods?
3.	About Squared ReLU and Separable Conv plus multi-head attention, I have privately conduct translation experiments on the original Transformer via replacing ReLU with Squared ReLU, and do not observe any improvement in bleu (maybe it is more suitable for the searched architecture or requires different specific hyper-parameters). And for Separable Conv plus multi-head attention has been used in many previous works, including NAS-based methods [a][b][c] and some computer vision works [d][e] also point out that Separable Conv plus multi-head attention can bring additional benefits. Therefore, it is unnecessary to emphasize the improvement is from Squared ReLU and Separable Conv plus multi-head attention in the search space. And more importantly, a convincing approach to validate the effectiveness of Squared ReLU and Separable Conv plus multi-head attention is using a search space with the constraints that we can not use Squared ReLU and Separable Conv plus multi-head attention. In other words,  under these constraints, maybe we can get other advanced architectures, rather than Squared ReLU and Separable Conv plus multi-head attention. However, the authors only tried to replace the corresponding module on the original Transformer and Primer in the ablation study, rather than replace the search space and re-search architecture.
4.	Will the searching code be released?


[a] AdaBERT: Task-Adaptive BERT Compression with Differentiable Neural Architecture Search.

[b] NAS-BERT: Task-Agnostic and Adaptive-Size BERT Compression with Neural Architecture Search

[c] AutoTinyBERT: Automatic Hyper-parameter Optimization for Efficient Pre-trained Language Models.

[d] CvT: Introducing Convolutions to Vision Transformers

[e] Co-Scale Conv-Attentional Image Transformers



**Time Spent Reviewing:**

12

---

> ### Author Response · Authors · 2021-08-10
> **Response to Reviewer dssT**
>
> Thank you for taking the time to review our work and provide feedback. We respond to quoted comments below:
>
> \
> \
> *“The contribution of this paper is incremental ... the searching algorithm is based on the existing works (Evolutionary search) ... as a NAS method, the searching algorithm is more important than the searched model. However, the authors cost many columns for introducing the searched model.”*
>
> The main contribution of this paper is not the search algorithm. Our main contribution is a low level open-ended search space and, most importantly, the resulting Primer model. Our work is most similar to EfficientNet: using architecture search to identify an architecture that can be used by practitioners. EfficientNet (https://arxiv.org/abs/1905.11946) for example is widely used and highly cited: ~3200 citations after ~2 years. Considering the impact of Transformer, if the resulting Primer model can be used in place of Transformer to save compute and energy, it would also be impactful and thus an important contribution to the field.
>
> \
> \
> *“This paper argues the improvements are mainly from Squared ReLU and Multi-DConv-Head Attention. So these modules are determined by neural architecture search, or just hand-crafted design?”*
>
> These modules were found by the architecture search.
>
> The search method first identified the Primer architecture. We then did an ablation study on each modification one at a time and found two modifications that are useful; we refer to them as Squared ReLU and Multi-DConv-Head Attention. Such ablation studies can be automated quite easily by identifying the modifications, and testing them out one at a time. In summary, we did not hand-craft Squared ReLU and Multi-DConv-Head Attention, and the decision to keep them in Primer-EZ was done by an ablation study that can be easily automated.
>
> Note, both Primer (unprocessed, found directly by architecture search) and Primer-EZ (post-processed by the ablation study) deliver similar results in T5. The raw search model (Primer) performs well without any modification. We develop Primer-EZ to make the results of our search easier to use, as a practical technique for the community. Please see Reviewer WYyB‘s comments on the difficulty of architecture adoption, which is the problem we are trying to address with Primer-EZ.
>
> \
> \
> *“Is Tensorflow Program necessary? If the core contributions of this paper are the searching algorithm and the searched models, I think it should not be sensitive to the code platform (like PyTorch or MXNet). In other words, I think it is unnecessary to highlight Tensorflow usage in a research paper.”*
>
> Tensorflow is not necessary. We include it to demonstrate the simplicity of our primitives vocabulary in a well established deep learning library. The discovered Primer modules can also be used in PyTorch, MXNet and other libraries.
>
> For example, the Tensorflow primitives vocabulary in Table 2 can be easily mapped to corresponding PyTorch primitives, as they are common low-level operations, such as SIN, ADDITION, CONV, etc.
>
>
> \
> \
> *“this paper aims to obtain a trade-off between the training time and sample efficiency, but what is the optimal trade-off has not been discussed.”*
>
> In this work, the tradeoff is determined by the search algorithm, which directly tries to find the highest accuracy model given a fixed training budget.
>
> Previous works have explicitly included efficiency metrics in fitness/reward functions for architecture search. For example, most are variants of the MNasNet objective (https://arxiv.org/abs/1807.11626), which explicitly considers model step time. However, these explicit objectives are often hard to tune, as one needs to manually decide the tradeoff between step time and task accuracy using hyperparameters (https://arxiv.org/abs/2008.06120). In this work, we circumvent the need for tuning such hyperparameters by having training efficiency included implicitly in our objective by restricting training compute budgets for models during the search. Thus, we do not manually define the optimal tradeoff, but rather let the search determine the tradeoff itself.
>
> \
> \
> *“Authors highlight that the multi-DConv-Head attention is important, but no ablation study has been conducted for analyzing the impact of this module.”*
>
> We explicitly conducted ablation experiments in the paper to analyze the impact of Multi-DConv-Head attention. See Appendix A.7 “Ablation and Insertion Studies.”
>
> We also mention this result in the main text of the paper. From page 4, line 148 of our paper: “we perform ablation tests across two codebases (T5 and Tensor2Tensor) and determine which of its modifications are generally useful (Appendix Figure 25). The two that produce the most robust improvements are squaring feed forward ReLUs and adding depthwise convolution to attention multi-head projections.”
>
> \
> \
> *“The part of the introduction section is like a program instruction. I am not sure whether it is appropriate for a research paper.”*
>
> Section 1 “Introduction” does not contain any program instructions.
>
> In Figures 1, 2, and 4 we include snippets of both Tensorflow code and pseudo-code. This is meant to represent the concepts in a familiar way for readers and make the ideas of the paper accessible to a wide audience.
>
> If the reviewer feels strongly about this, we can alter the figures to contain less code.

---

> ### Author Response · Authors · 2021-08-27
> **Second Response to Reviewer dssT**
>
> Thank you for taking the time to read our responses and offer further feedback. We respond to quoted open issues below:
>
> \
> \
> *'About Squared ReLU and Separable Conv plus multi-head attention, I have privately conduct translation experiments on the original Transformer via replacing ReLU with Squared ReLU, and do not observe any improvement in bleu...’*
>
> It is hard to provide feedback on your experiments without additional information about your configuration. We are open sourcing our T5 comparisons so that our results are exactly reproducible.
>
> Furthermore, in our Limitations Section we clearly state that our focus is on decoder-only models, and that transfer to encoder-decoder models (ex, translation) may be limited. Reviewers LGXf and WYyB both note this limitation and agree that a decoder-only study is a suitable scope for this work.
>
> Although Transformer is now used for a variety of purposes, its application to decoder-only autoregressive language modeling remains important to the community. At NeurIPS 2020, the Best Paper Award was given to GPT-3 (https://arxiv.org/pdf/2005.14165.pdf), which focused only on this style of autoregressive language modeling, demonstrating how it can be expanded to the pretraining and one-shot downstream task regime. To further demonstrate how Primer can benefit the community, we have conducted additional experiments that mimic the GPT-3 setup. GPT-3 was not open sourced, and so this is not an exact replication, but we have done our best to reproduce the results using a proprietary pretraining dataset and the same one-shot downstream tasks. The results of a 1.9B parameter comparison between Primer and Transformer+GELU in Lingvo are presented below. Note, these are *not* directly comparable to GPT-3, as our configuration is different, but we provide the GPT-3 numbers as well to demonstrate that our results are reasonable for the given model sizes. We train Transformer (at full compute) for 1M steps so that it is roughly on par with the performance of GPT-3 XL.
>
> Here we present crude aggregates that summarize the high level trend that Primer can achieve the same results as Transformer+GELU using ⅓ of the pretraining compute and better results when given the same pretraining compute: (See followup responses to reviewers LGXf and WYyB for full numbers, as they asked about pretraining performance).
>
> |                             | Transformer 1/3 Compute | Transformer Full Compute | Primer 1/3 Compute | Primer Full Compute | GPT-3 XL (Brown et al., 2020) |
> |-----------------------------|:-----------------------:|:------------------------:|:------------------:|:-------------------:|:-----------------------------:|
> |     Pretraining Val PPLX    |           15.3          |           14.3           |        14.3        |         13.5        |               -               |
> |    Mean QA One-Shot Score   |           30.9          |           34.5           |        34.6        |         36.8        |              34.3             |
> | Mean Multi-Choice One-Shot Score |           53.1          |           54.7           |         55         |         56.2        |              54.1             |
>
> Thus, although we limit the scope of our work to decoder-only autoregressive language modeling, we do feel that Primer can have positive community impact, especially as large language model training continues to grow in size with the increasing popularity of pretraining.
>
> \
> *‘...(maybe it is more suitable for the searched architecture or requires different specific hyper-parameters).’*
>
> We demonstrate our results transfer to three independent code bases (T2T, T5 and Lingvo) using the existing Transformer hyperparameters without *any* additional tuning. However, in all cases we focus only on autoregressive language modeling and do not make claims in the paper that methods transfer beyond that (such as to encoder-decoder translation).
>
> It is hard to provide additional feedback on your experiments without additional information about your configuration. We are open sourcing our T5 comparisons so that our results are exactly reproducible.
>
> \
> \
> \
> *‘Searching efficient transformer has been surveyed in many previous works [a][b][c]. The difference between these works is that they only focus on lightweight compressed transformers due to the limitation of resources … a question is if under the same computation, is it possible to obtain a better model than other NAS methods?’*
>
> The search method is not the focus of this work and for that reason other search methods may perform better than Regularized Evolution. If future works investigate the application of additional search methods in our space and find even stronger models, that will further confirm that the low level search we proposed is a promising direction for NAS moving forward.
>
> \
> \
> \
> *‘Just as aforementioned, your paper focuses more attention on the model (named Primer) rather than the searching algorithm. Therefore, I think your title needs to be changed, like "Primer: xxx" rather than "searching for efficient transformer". Otherwise, it is unfair for other NAS papers, let alone you have not compared with any other NAS-based methods. And your paper is mainly focused on the autoregressive language model, called "for language modeling", which is also a little over-claim.’*
>
> We are happy to add ‘Primer’ to the title, to emphasize its importance in this work: ‘Primer: Searching for Efficient Transformers for Language Modeling.’ We feel that within the ML community, ‘language modeling’ implicitly means autoregressive language modeling, and so our title uses that for succinctness. In the abstract and throughout the paper, we clearly note that we are focused on autoregressive language modeling and also admit this in our Limitations section. We are happy to add ‘autoregressive’ to the title if you feel that it will improve the work.
>
> \
> \
> \
> *‘And for Separable Conv plus multi-head attention has been used in many previous works, including NAS-based methods [a][b][c] and some computer vision works [d][e] also point out that Separable Conv plus multi-head attention can bring additional benefits.’*
>
> Our work is differentiated from these NAS-based methods ([a,b,c]) in that we discover the utility of applying convolution *within* the attention mechanism, whereas these other works apply attention and convolution as separate layers. What’s more, these works focus on BERT, not autoregressive language modeling.
>
> We note the similarity to vision architectures in Section 3, with proper citations (ex, [d]), but also highlight the differences there as well. Our discovered “Multi-DConv-Head Attention” has never been proposed, to the best of our knowledge.
>
> Additionally, we use Depthwise Conv, not Separable Conv. Depthwise Conv is more compute efficient and compute efficiency is the focus of this study. We additionally compare to Separable Conv and find that it is not as effective (see Appendix A6, mentioned in Section 3 of the main text).
>
> \
> \
> \
> *‘It is unnecessary to emphasize the improvement is from Squared ReLU and Separable Conv plus multi-head attention in the search space. And more importantly, a convincing approach to validate the effectiveness of Squared ReLU and Separable Conv plus multi-head attention is using a search space with the constraints that we can not use Squared ReLU and Separable Conv plus multi-head attention. In other words, under these constraints, maybe we can get other advanced architectures, rather than Squared ReLU and Separable Conv plus multi-head attention…’*
>
> Separable Conv and Squared ReLU are *not* building blocks in our search space. Our search space is differentiated from previous NAS works in that Multi-DConv-Head Attention and Squared ReLU are discoverable, not that they are ingredients in our search vocabulary. We cannot remove them from the search space without removing the simple primitives that define them. For example, to remove Squared ReLU, as proposed, would require removing multiplication from our space, meaning we cannot even represent the original Transformer.
>
> \
> *‘...However, the authors only tried to replace the corresponding module on the original Transformer and Primer in the ablation study, rather than replace the search space and re-search architecture.’*
>
> These searches are expensive to run and thus cannot be trivially rerun for the sake of ablating small portions of the search space.
>
> The process of ‘tuning’ search spaces via ablation has been used in NAS before, but increases manual burden by requiring hand-crafting the ablations and severely biases the outcome of these searches (https://arxiv.org/abs/2008.06120). Our search space is specifically constructed to move away from that paradigm. Instead of extensive hand-tuning via ablation, we use an open-ended search space of simple primitives (Table 3), and let the search determine what is useful.
>
> Additionally, the modules of our ablation study were those found by the search. These modules are not hand-crafted building blocks for the search space. We cannot remove them from the search space without removing the simple primitives that define them. For example, to remove Squared ReLU, as proposed, would require removing multiplication from our space, meaning we cannot even represent the original Transformer.
>
> \
> \
> \
> *‘Will the searching code be released?’*
>
> The search code is deeply integrated with proprietary distributed compute infrastructure but we will investigate if it’s possible to release it. We will release code to reproduce our comparisons in T5.

---

### Official Review · Reviewer_rJ28 · 2021-07-19

**Rating:** 7
**Confidence:** 4

**Summary:**

The paper defines a new neural architecture search framework based on generating TensorFlow-based neural network programs. The paper discusses the search space and the evolutionary search-based searching method. The authors perform the search algorithm on a language modeling benchmark and show two specific modifications to the original Transformer model in the resulting model generalizes well to other tasks: squaring ReLU activation and adding a spatial depth-wise convolution after the QKV transformation of the multi-head attention layer.



**Limitations And Societal Impact:**

Please refer to the main review section.



**Main Review:**

I lean towards accepting the paper. The paper performs neural architecture search on a much larger searching space compared to previous works and provides insights from search results that would be useful for a much wider set of language modeling tasks. Detailed comments:

Strengths:
- This paper works on a much larger search space compared to previous works.
- The distilled insights (i.e. two techniques in Primer-EZ) would be greatly beneficial for the community. I believe this actually will ease the reproduction of the paper because it can now be verified by the evaluation of the effectiveness of these two techniques.
- The evaluation in the paper is comprehensive and the paper is clear and easy to understand.

Weaknesses:
- It would be better to include deeper discussions on why the resulting modification performs well.
- The search result is only a slight modification of the original transformer structure. This seems to contradict the starting point of the paper, which is to introduce a larger search space compared to previous works.

Minor issues:
- Line 59 “a evolutionary search DNA” -> “an evolutionary”


**Time Spent Reviewing:**

3

---

> ### Author Response · Authors · 2021-08-10
> **Response to Reviewer rJ28**
>
> Thank you for taking the time to review our work and provide feedback. We respond to quoted comments below:
>
> \
> \
> *“The search result is only a slight modification of the original transformer structure. This seems to contradict the starting point of the paper, which is to introduce a larger search space compared to previous works.”*
>
> The open-ended Primer search space is meant to expand the types of architecture modifications that can be discovered and remove bias from the search.
>
> Previous architecture search works that have yielded practical results rely heavily on well-designed search spaces composed of powerful high-level building blocks. These search spaces are injected with a significant amount of human bias, such that random sampling produces high quality models (see https://arxiv.org/abs/1902.07638 https://arxiv.org/abs/1902.08142 and https://arxiv.org/abs/2008.06120 ).
>
> The Primer search space is much more open-ended. The downside is that many of its architectures are completely degenerate. For example, ~78% of randomly sampled models cannot train for more than five minutes due to numerical instability. The upside is that modifications can be made at arbitrary locations in the modeling and a wider variety of architectures can be found. For instance, Multi-Dconv-Head Attention changes the self-attention transformation itself, rather than merely using self-attention as a search ingredient as previous works have (https://arxiv.org/abs/1901.11117). In this sense, although Mutli-Dconv-Head Attention is a simple modification, it does utilize the granularity of our open-ended search space.
>
> A limitation of our work is the number of candidate architectures we can evaluate, given the high compute cost of each evaluation. This is why we use conceptual initialization -- because the cost of evaluating weak models is high and our search space has many weak models. Future work can explore more efficient algorithms that are able to better explore the Primer space and potentially find models that are even more different from Transformer.
>
> \
> \
> *“It would be better to include deeper discussions on why the resulting modification performs well.”*
>
> We will do further analysis of the learned self-attention patterns and depthwise convolution kernels and include any interesting findings in the next iteration of the paper.

---

### Official Review · Reviewer_LGXf · 2021-07-19

**Rating:** 9
**Confidence:** 4

**Summary:**

This work performs a neural architecture search to find an alternative to Transformer language models for efficient training and inference. Primer is a resulting architecture and its two main modifications are (1) squaring ReLU activations and (2) adding depthwise convolution to attention multi-head projections. Primer and Primer-EZ (applied these two modifications only) achieve better efficiency-accuracy trade-off while also following a scaling law like Transformer.

**Limitations And Societal Impact:**

The paper honesty reveals its limitations and suggests meaningful future directions. This work focuses on autoregressive language modeling (decoder-only). Nevertheless, Appendix A.11 includes results on encoder-decoder masked language modeling. Results on encoder-only language models like BERT would be also interesting. Since Transformer is universally used for encoder-only, decoder-only, and encoder-decoder, I am curious whether Primer can take a similar role.

I am wondering Primer can perform well even on a GPT-3 scale as the authors mentioned in the limitation section and other modalities.

Does the proposed method generalize to other deep learning libraries other than TensorFlow like PyTorch?

**Main Review:**

The paper is well-organized and clearly written. Extensive experimental results and analysis on them support the effectiveness of the proposed method. Although the scope of this work is for autoregressive language modeling, the impact of this work could be quite significant in that Primer can be used for the alternative to Transformer which is currently a defacto standard architecture for NLP and many other ML problems. Primer is found by the automatic architecture search rather than designed by human intuition

Strong initialization with Transformer what the authors called conceptual initialization is a neat idea for the efficient search. However, it reduces the scope of possible architecture significantly, limiting a chance to find a largely different novel architecture. It would be nice if there is a way to alleviate this hard limit with some kind of relaxation.

It would be interesting to check whether the search space contains previous works on modifying a Transformer architecture in near search steps from the initialization after an exhaustive literature survey on those works or fix the search strategy inspired by those methods.

It is questionable that the most training efficient architecture (having the best training accuracy with a fixed amount of budget) is also the most accurate model having the best test accuracy because generalization ability might be sacrificed.

Is there any way to include efficiency metrics directly to a fitness score during the search?

In Section 3, other modifications of Primer except two major modifications are removed by future post-processing. Is there a way to prevent these marginal or not generalizable modifications as a result of the search in advance without requiring human intervention?

Minor: the term of DNA is used without mentioning the unabbreviated version (maybe Differential Neural Architecture) first.

**Time Spent Reviewing:**

12 hours

---

> ### Author Response · Authors · 2021-08-10
> **Response to Reviewer LGXf**
>
> Thank you for taking the time to review our work and provide feedback. We respond to quoted comments below:
>
> \
> \
> *“I am wondering Primer can perform well even on a GPT-3 scale as the authors mentioned in the limitation section and other modalities.”*
>
> We intend on releasing additional results that demonstrate Primer’s gains hold at a 1B parameter size under training conditions similar to GPT-3. Specifically, we find Primer’s improvements on pretraining perplexity transfer to downstream one-shot tasks as well.
>
> \
> \
> *“Strong initialization with Transformer what the authors called conceptual initialization is a neat idea for the efficient search. However, it reduces the scope of possible architecture significantly, limiting a chance to find a largely different novel architecture. It would be nice if there is a way to alleviate this hard limit with some kind of relaxation.”*
>
> Conceptual initialization is not actually a hard limit, in the sense that it does not limit the size of the search space if the number of search individuals is not bounded. As the evolutionary search is conducted, the population can drift arbitrarily far from the initialization seed (Transformer) if given enough compute to create an adequate number of generations. Thus, with enough compute, any model in our open ended space is discoverable, even with conceptual initialization.
>
> In our particular search setup, the novelty of the discovered model is limited by the number of candidates we have the compute to explore. We only have the resources to consider ~25K individuals using a population size of 100, meaning that at most the discovered model can only be ~250 mutations away from the original Transformer.
>
> In future work, approaches that drastically reduce the compute needed to evaluate each search candidate (such as https://arxiv.org/abs/1806.09055) could be applied to the Primer search space in conjunction with conceptual initialization to venture farther from the initialization point and perhaps yield even stronger and more novel architectures.
>
> \
> \
> *“In Section 3, other modifications of Primer except two major modifications are removed by future post-processing. Is there a way to prevent these marginal or not generalizable modifications as a result of the search in advance without requiring human intervention?”*
>
> Yes. We did a simple ablation study using each modification one at a time to determine what is most useful. Such ablation studies can be automated quite easily by identifying the modifications, and testing them out one at a time. We did not, for instance, hand-select the combination that we thought was best using our own intuitions.
>
> \
> \
> *“Does the proposed method generalize to other deep learning libraries other than TensorFlow like PyTorch?”*
>
> Our search can be trivially applied on top of other deep learning libraries, such as PyTorch. For example, the Tensorflow primitives vocabulary in Table 2 can be easily mapped to corresponding PyTorch primitives, as they are common low-level operations, such as SIN, ADDITION, CONV, etc. We include the Tensorflow details in our paper to demonstrate the simplicity of our primitives vocabulary in a well established deep learning library.
>
> In terms of Primer’s quality vs. train steps, we believe that these results should transfer to any platform that implements the same training configuration. For this reason, we will open source our T5 experiments so that they can be tested and copied by anyone who wishes to reimplement our results.
>
> In terms of Primer’s compute saved, the saving factors may vary depending on the efficiency of other libraries’ implementations of specific operations. For instance, Primer uses depthwise convolution. If a library has a relatively slower implementation of depthwise convolution, the savings may be worse. If a library has a relatively faster implementation of depthwise convolution, the savings may be better.
>
> \
> \
> *“Is there any way to include efficiency metrics directly to a fitness score during the search?”*
>
> Previous works have explicitly included efficiency metrics in fitness/reward functions for architecture search. For example, most are variants of the MNasNet objective (https://arxiv.org/abs/1807.11626), which explicitly considers model step time. However, these explicit objectives are often hard to tune, as one needs to manually decide the tradeoff between step time and task accuracy using hyperparameters (https://arxiv.org/abs/2008.06120). In this work, we circumvent the need for tuning such hyperparameters by having training efficiency included implicitly in our objective by restricting training compute budgets for models during the search.
>
> \
> \
> *“It would be interesting to check whether the search space contains previous works on modifying a Transformer architecture in near search steps from the initialization after an exhaustive literature survey on those works or fix the search strategy inspired by those methods.”*
>
> This is a great idea for future work! We tried to make the search space as open ended and flexible as possible, but this type of exploration would surely point us to relevant techniques that we do not currently cover.
>
> \
> \
> *“It is questionable that the most training efficient architecture (having the best training accuracy with a fixed amount of budget) is also the most accurate model having the best test accuracy because generalization ability might be sacrificed.”*
>
> We will improve our descriptions in the paper to clarify that we measure training-time performance using validation accuracy, not training accuracy, to mitigate generalization issues. In our experiments, during training, each model is regularly evaluated on the validation set so that we have a history of validation accuracy vs. number of train steps (or training compute hours).
>
> \
> \
> *“Results on encoder-only language models like BERT would be also interesting.”*
>
> We currently find that the application to models using encoders is limited. Future work could explore a similar style of search aimed at improving encoder-decoder or encoder-only models.

---

> ### Author Response · Authors · 2021-08-27
> **Second Response to Reviewer LGXf  (One-shot Results)**
>
> As noted in our first response, we have conducted additional experiments that mimic the one-shot GPT-3 setup (https://arxiv.org/pdf/2005.14165.pdf). GPT-3 was not open sourced, and so this is not an exact replication, but we have done our best to reproduce the results using a proprietary pretraining dataset and the same one-shot downstream tasks. The results of a 1.9B parameter comparison between Primer and Transformer+GELU (approximating the GPT-3 architecture) in Lingvo are presented below. Note, these are *not* directly comparable to GPT-3, as our configuration is different, but we provide the GPT-3 numbers as well to demonstrate that our results are reasonable for the given model sizes. We train Transformer (at full compute) for 1M steps so that it is roughly on par with the performance of GPT-3 XL.
>
> \
> First, we present crude aggregates that summarize the high level trend that Primer can achieve the same results as Transformer+GELU using ⅓ of the pretraining compute and better results when given the same pretraining compute:
>
> |                             | Transformer 1/3 Compute | Transformer Full Compute | Primer 1/3 Compute | Primer Full Compute | GPT-3 XL (Brown et al., 2020) |
> |-----------------------------|:-----------------------:|:------------------------:|:------------------:|:-------------------:|:-----------------------------:|
> |     Pretraining Val PPLX    |           15.3          |           14.3           |        14.3        |         13.5        |               -               |
> |    Mean QA One-Shot Score   |           30.9          |           34.5           |        34.6        |         36.8        |              34.3             |
> | Mean Multi-Choice One-Shot Score |           53.1          |           54.7           |         55         |         56.2        |              54.1             |
>
> \
> \
> We now present the exact scores for each task:
>
> |                 | Transformer 1/3 Compute | Transformer Full Compute | Primer 1/3 Compute | Primer Full Compute | GPT-3 XL (Brown et al., 2020) |
> |-----------------|:-----------------------:|:------------------------:|:------------------:|:-------------------:|:-----------------------------:|
> |     TriviaQA    |    % 22.49 +/- 0.37 %   |      26.81 +/- 0.45      |   27.50 +/- 0.38   | # 32.16 +/- 0.47 #|              26.5             |
> |      WebQs      |      9.06 +/- 0.50      |       9.62 +/- 0.42      |    9.84 +/- 0.81   |    10.44 +/- 0.34   |              9.15             |
> |       NQs       |    % 5.84 +/- 0.21 %    |       6.73 +/- 0.19      |  # 7.83 +/- 0.50 # |  # 9.08 +/- 0.30 #  |              5.43             |
> |     SQuADv2     |    % 54.15 +/- 2.36 %   |      65.43 +/- 2.86      |   64.15 +/- 3.65   |    67.81 +/- 1.15   |               54              |
> |     LAMBADA     |    % 51.49 +/- 0.94 %   |      55.20 +/- 1.30      |   54.52 +/- 1.14   |    56.81 +/- 0.94   |              58.3             |
> |       CoQa      |    % 52.54 +/- 1.13 %   |      57.74 +/- 1.18      |   59.06 +/- 0.87   |  # 61.18 +/- 0.65 # |              66.1             |
> |       DROP      |    % 21.46 +/- 0.44 %   |      23.35 +/- 0.22      | # 24.83 +/- 0.50 # |  # 26.46 +/- 0.24 # |               23              |
> |       Quac      |      30.06 +/- 0.54     |      30.85 +/- 0.67      |   28.91 +/- 0.93   |    30.22 +/- 0.65   |              32.3             |
> |                 |                         |                          |                    |                     |                               |
> |    HellaSwag    |    % 55.73 +/- 0.26 %   |      59.45 +/- 0.22      | # 60.18 +/- 0.27 # |  # 63.26 +/- 0.20 # |              53.5             |
> |    StoryCloze   |      75.18 +/- 0.25     |      75.92 +/- 0.43      | # 76.85 +/- 0.19 # |  # 77.46 +/- 0.34 # |              74.2             |
> |    Winogrande   |    % 55.35 +/- 0.27 %   |      58.37 +/- 0.43      |   58.82 +/- 0.30   |  # 60.39 +/- 0.22 # |              59.1             |
> |       PIQA      |      72.59 +/- 0.45     |      72.64 +/- 0.28      |   73.65 +/- 0.54   |  # 74.97 +/- 0.35 # |              74.4             |
> | ARC (Challenge) |    % 32.65 +/- 0.39 %   |      34.39 +/- 0.33      |   35.58 +/- 0.92   |  # 37.36 +/- 0.36 # |              36.4             |
> |    ARC (Easy)   |      64.53 +/- 0.49     |      64.87 +/- 0.47      |   65.60 +/- 0.62   |  # 67.49 +/- 0.53 # |              55.9             |
> |     ANLI R1     |      33.93 +/- 1.22     |      35.48 +/- 0.24      |   35.48 +/- 0.38   | % 34.84 +/- 0.25 %  |              34.6             |
> |     ANLI R2     |      33.47 +/- 0.72     |      33.42 +/- 0.53      |   34.46 +/- 0.63   |    33.50 +/- 0.37   |              32.7             |
> |     ANLI R3     |      34.49 +/- 0.71     |      35.22 +/- 0.13      | % 33.00 +/- 0.26 % |  % 33.80 +/- 0.47 % |              33.9             |
> |      ReCoRD     |    % 84.78 +/- 0.07 %   |      86.29 +/- 0.16      |   85.84 +/- 0.26   |  # 86.68 +/- 0.04 # |               83              |
> |       WSC       |      67.40 +/- 0.84     |      66.79 +/- 1.22      |   69.30 +/- 1.25   |    68.86 +/- 1.23   |              62.5             |
> |      BoolQ      |    % 58.89 +/- 1.14 %   |      63.63 +/- 2.12      |   60.72 +/- 0.80   |    64.73 +/- 1.98   |              63.7             |
> |        CB       |      56.25 +/- 2.46     |      52.98 +/- 2.69      |   55.36 +/- 3.34   |    56.55 +/- 9.63   |              48.2             |
> |       RTE       |    % 48.38 +/- 1.19 %   |      53.55 +/- 2.50      |   54.30 +/- 1.54   |    52.89 +/- 2.81   |              49.5             |
> |       COPA      |    % 80.17 +/- 3.19 %   |      87.17 +/- 1.17      |   84.80 +/- 1.48   |    87.50 +/- 1.05   |               74              |
> |       WiC       |      51.59 +/- 0.23     |      50.97 +/- 0.48      | # 51.69 +/- 0.07 # |  # 51.80 +/- 0.09 # |              49.2             |
> |    OpenBookQA   |      45.33 +/- 0.88     |      46.83 +/- 0.75      |   47.92 +/- 0.41   |  # 49.30 +/- 0.45 # |              46.4             |
> |      RACE-h     |    % 39.44 +/- 0.43 %   |      40.80 +/- 0.36      |   40.40 +/- 0.35   |  # 43.71 +/- 0.33 # |               42              |
> |      RACE-m     |    % 49.95 +/- 1.02 %   |      52.56 +/- 0.41      |   51.79 +/- 0.80   |  # 54.00 +/- 0.42 # |              55.2             |
>
> Sample means and standard deviations are computed using the evaluated performance of 5 weight checkpoints.
>
> \# denotes improved performance over Transformer with full compute that is statistically significant under independent two-sample t-test with p-value threshold 0.05.
>
> % denotes worse performance than Transformer with full compute that is statistically significant under independent two-sample t-test with p-value threshold 0.05.
>
> \
> Primer with ⅓ compute achieves equal performance with Transformer using full compute on 21 tasks, better performance on 5 tasks and worse performance on 1 task. Primer with full compute achieves equal performance with Transformer using full compute on 10 tasks, better performance on 15 tasks and worse performance on 2 tasks.
>
> Thus, we believe the benefits of Primer extend beyond pure language modeling, and can be applied to the pretraining/one-shot regime. This can increase Primer's community impact, especially as large language model training continues to grow in size with the increasing popularity of pretraining.

---

### Decision · Program_Chairs · 2021-09-27

**Decision:**

Accept (Poster)

**Comment:**

The submission makes two main contributions: defining neural architecture search using lower level primitives than previous work, and using this search to introduce a Transformer variant they call Primer. Results show that Primer can outperform vanilla Transformers on various language modeling tasks.

While three of the reviewers support acceptance, Reviewer dssT raises several concerns, arguing that a new search space is not enough of a contribution. The reviewer also would like to see a search space that does not include the squared ReLU and Separable conv plus multi-head attention that are used in the final solution. Other reviewers appreciated the larger, low level search space. I think that the authors convincingly rebut the need to exclude modules from the search space, as for example, excluding squared ReLU would require excluding multiplication. There are also concerns that it is only evaluated on language modeling, but the authors are up front about this limitation, and I agree with most of the reviewers that this is an important and general enough task to focus on.

Overall, the consensus is that this is a good paper that should be accepted.